# CO$_2$/carbonate-mediated electrochemical water oxidation to hydrogen peroxide

Lei Fan[1,2,7], Xiaowan Bai [3,7], Chuan Xia [1,4,7], Xiao Zhang [1], Xunhua Zhao [3], Yang Xia[1], Zhen-Yu Wu [1], Yingying Lu [2✉], Yuanyue Liu [3✉] & Haotian Wang [1,5,6✉]

Electrochemical water oxidation reaction (WOR) to hydrogen peroxide (H$_2$O$_2$) via a 2e$^-$ pathway provides a sustainable H$_2$O$_2$ synthetic route, but is challenged by the traditional 4e$^-$ counterpart of oxygen evolution. Here we report a CO$_2$/carbonate mediation approach to steering the WOR pathway from 4e$^-$ to 2e$^-$. Using fluorine-doped tin oxide electrode in carbonate solutions, we achieved high H$_2$O$_2$ selectivity of up to 87%, and delivered unprecedented H$_2$O$_2$ partial currents of up to 1.3 A cm$^{-2}$, which represents orders of magnitude improvement compared to literature. Molecular dynamics simulations, coupled with electron paramagnetic resonance and isotope labeling experiments, suggested that carbonate mediates the WOR pathway to H$_2$O$_2$ through the formation of carbonate radical and percarbonate intermediates. The high selectivity, industrial-relevant activity, and good durability open up practical opportunities for delocalized H$_2$O$_2$ production.

[1] Department of Chemical and Biomolecular Engineering, Rice University, Houston, TX 77005, USA. [2] State Key Laboratory of Chemical Engineering, Institute of Pharmaceutical Engineering, College of Chemical and Biological Engineering, Zhejiang University, Hangzhou 310027, China. [3] Texas Materials Institute and Department of Mechanical Engineering, The University of Texas at Austin, Austin, TX 78712, USA. [4] Smalley-Curl Institute, Rice University, Houston, TX 77005, USA. [5] Department of Materials Science and NanoEngineering, Rice University, Houston, TX 77005, USA. [6] Department of Chemistry, Rice University, Houston, TX 77005, USA. [7] These authors contributed equally: Lei Fan, Xiaowan Bai, Chuan Xia. ✉email: yingyinglu@zju.edu.cn; Yuanyue.liu@austin.utexas.edu; htwang@rice.edu

Electrochemical water ($H_2O$) oxidation to hydrogen peroxide ($H_2O_2$) via a $2e^-$ pathway ($2e^-$-WOR) represents a green and sustainable route to produce $H_2O_2$ compared to traditional anthraquinone process, but is currently challenged by low selectivity and activity due to strong competition from the typical $4e^-$ oxygen evolution reaction pathway (OER or $4e^-$-WOR)[1–6]. Traditional approaches to promoting $2e^-$-WOR to $H_2O_2$ have been mostly focused on exploring catalysts with relatively weak binding strength with intermediate O species compared to that in the $4e^-$ counterpart[7–11]. These catalysts (typically made of inert metal oxides[7–10] as well as other materials[12–14]) requires large over-potentials to activate the water oxidation step, but their $2e^-$-WOR current densities are usually limited at ~10 to 200 mA cm$^{-2}$, as the extra overpotentials to drive larger currents would start to push the water oxidation reaction all the way down to $O_2$ with significantly decreased $H_2O_2$ selectivity[7–11,15,16]. As a result, the state-of-the-art $2e^-$-WOR performances are still far below the requirements in practical applications[7,17].

Reaction redox mediators as "electron shuttles" have been playing important roles in facilitating desired reaction pathways (especially for intermediate products) in electrocatalysis[18,19], electroorganic synthesis[20–22] and bioelectrocatalysis[23], and could become a solution to the $H_2O_2$ activity-selectivity dilemma. While exploring such kind of reaction mediators for selective $2e^-$-WOR is challenging, our nature may have an answer for us. $H_2O_2$ commonly exists in biosystems as one type of reactive oxygen species (ROS)[24] which are essential in signal transduction[25,26]. However, researchers have found out that high concentrations of $CO_2$ could cause ROS burst (rapid release of ROS from cells), leading to destructions of redox-sensitive proteins and cell structures (Supplementary Note 1)[27,28].

Inspired by this phenomenon and previous studies in electro-generated chemiluminescence[29,30] and peroxycarbonate synthesis[31], here we hypothesize that $CO_2$ (or carbonate as its ionic form in water) may serve as an effective mediator to promote the $H_2O_2$ pathway in electrochemical water oxidation (Fig. 1a)[32–34]. Using fluorine doped tin oxide (FTO) as a model catalyst electrode, we demonstrated high $H_2O_2$ selectivity up to 87%, delivered high $H_2O_2$ partial current densities up to 1.3 A cm$^{-2}$, and achieved a long-term stable and continuous $H_2O_2$ generation for 250 hours with over 80% $H_2O_2$ selectivity at 150 mA cm$^{-2}$ current density in carbonate solutions. The electrochemical performance of our work represents orders of magnitude improvement compared to previous works. We studied the mechanism of the carbonate mediations effects using molecular dynamics simulations, coupled with electron paramagnetic resonance and isotope labeling experiments. Simulation and experimental results suggested that carbonate mediates the WOR pathway through carbonate radical and percarbonate intermediates.

## Results and discussion

**Verification of the $CO_2$/carbonate mediation effects.** To support the $CO_2$/carbonate mediator strategy, an oxidation catalyst electrode which meets the following criteria is a prerequisite: first, it should be an inert catalyst for the $4e^-$ oxygen evolution reaction (OER); second, it has a high electrical conductivity to deliver high currents; third, it should remain stable under high oxidation potentials in water. After an initial screening process, FTO (Supplementary Fig. 1) was selected as the catalytic electrode compare to other materials (Supplementary Fig. 2). While FTO has also been used in some previous studies of photoelectrochemical or electrochemical water oxidation reaction, in most cases it was used as the substrate for studying other catalytic materials[7,8]. Additionally, the $H_2O_2$ selectivity and activity on FTO reported before are rather low[8], and the surface reaction

mechanism of $2e^-$-WOR is still unclear yet. We evaluated the electrochemical WOR performance using a standard three-electrode setup in an H-type cell. Sodium phosphate buffer (0.65 M $Na_2HPO_4$ and 0.35 M $NaH_2PO_4$, pH ~7) was chosen as the aqueous electrolyte, due to its high stability under oxidative potentials[35] and strong buffering capability[36], for investigating the possible impacts of $CO_2$ on WOR pathways (Fig. 1b and Supplementary Fig. 3). In Ar-saturated 1.0 M sodium phosphate buffer, only trace amount of $H_2O_2$ was detected within a wide range of applied water oxidation potentials, with $H_2O_2$ Faradaic efficiencies (FEs) less than 1% (Fig. 1b). This result suggests that FTO presents an intrinsic selectivity towards the $4e^-$-WOR pathway. Surprisingly, when the solution was saturated with $CO_2$, a significant jump of $H_2O_2$ selectivity was achieved under the same reaction conditions (Fig. 1b), supporting our hypothesis that $CO_2$ can play a role in steering the WOR reaction pathway towards $H_2O_2$.

As $CO_2$ exists in different types of species in aqueous solutions[37], including dissolved $CO_2$, carbonate, and bicarbonate, we therefore designed control experiments to better identify the key factors that are at play in promoting $H_2O_2$ generation. We first used 1.0 M $NaHCO_3$ as the electrolyte, where bicarbonate is the dominant species, to evaluate FTO's $2e^-$-WOR performance. As shown in Fig. 1c–e and Supplementary Fig. 4, the peak $H_2O_2$ FE was ~ 34%, corresponding to a $H_2O_2$ partial current density of only 5.6 mA cm$^{-2}$. We further switched the electrolyte to carbonate dominated 1.0 M $Na_2CO_3$ solution and performed the same test, and observed a drastic change. The $H_2O_2$ FE jumped up to a maximal of 56% at 3.2 V versus reversible hydrogen electrode (vs. RHE) with a significantly improved $H_2O_2$ partial current density of over 50 mA cm$^{-2}$ (Fig. 1e), representing over one order of magnitude increase compared to that in either $CO_2$ saturated electrolyte or bicarbonate electrolyte. To explore if this promotion effect on $2e^-$-WOR pathway is an intrinsic catalytic property of the FTO catalyst, or specifically exists in the $CO_2$/bicarbonate/carbonate systems, we also tested the $H_2O_2$ selectivity in other commonly used electrolytes containing different anion species, including sodium sulfate, sodium nitrate, sodium hydroxide, sodium perchloride, as well as the sodium phosphate buffer we showed earlier (Fig. 2a–d and Supplementary Fig. 5). To further confirm the carbonate promotion effects, we then tested the electrochemical water oxidation performance in $Na_2CO_3$ + NaOH electrolyte (Supplementary Fig. 6a). In alkaline electrolytes, $OH^-$ absorption plays a key role in oxygen evolution reaction[38]. With the increase of NaOH concentration, $OH^-$ absorption will decrease the coverage ratio of carbonate absorption on FTO surface, and thus leading to decreased $H_2O_2$ selectivity and increased OER, which further demonstrates that high pH is not the reason for high $H_2O_2$ selectivity when compared to bicarbonate solutions. As a result, none of them presented any preferences for $2e^-$-WOR pathway, with negligible $H_2O_2$ selectivity of less than 2%, which indicates the participation of $CO_2$-related species in the electrochemical $H_2O$-to-$H_2O_2$ conversion process on the FTO surface.

Since in carbonate solution the FTO catalyst exhibited the best $H_2O_2$ generation performance, we thus tested the dependence of $H_2O_2$ selectivity over $Na_2CO_3$ concentration to further reveal the promotion effects of carbonate on $2e^-$-WOR. As shown in Fig. 2e, the $H_2O_2$ FE under 100 mA cm$^{-2}$ current presented a monotonic enhancement from 5% to 56% with increased $Na_2CO_3$ concentration from 0.1 M to 2.0 M, respectively, indicating the critical role of carbonate ions in promoting $H_2O_2$ selectivity. Possible promotion effects from sodium ions were excluded as the $H_2O_2$ FEs were similar under different $Na^+$ concentrations (Fig. 2 f). We also examined the $O_2$ FEs from the $4e^-$-WOR pathway using gas chromatography quantification, which added

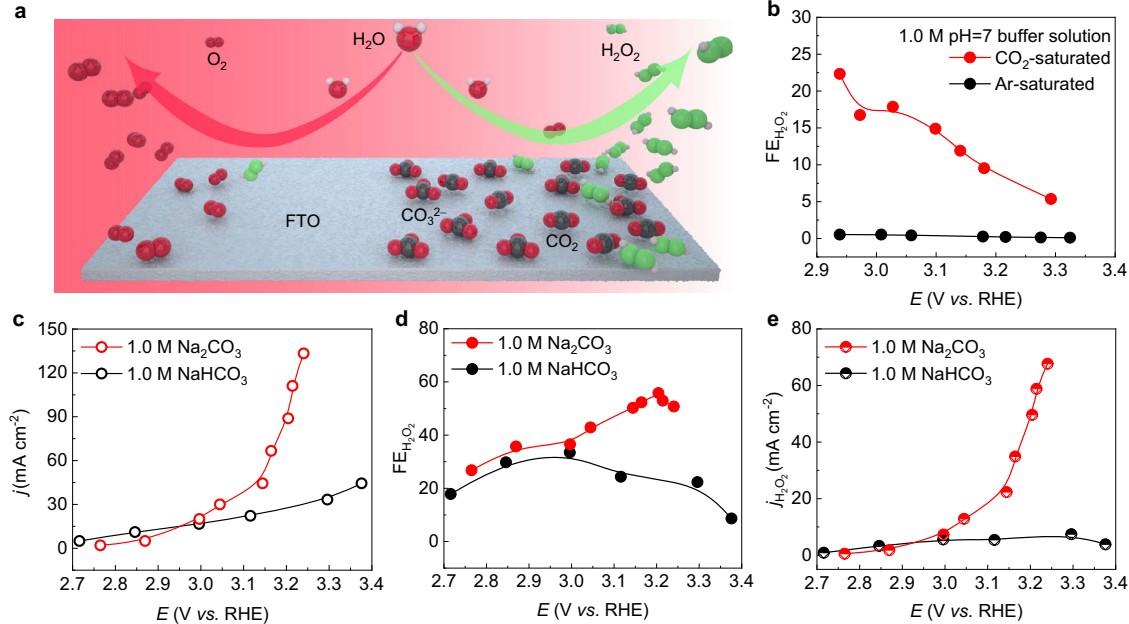

**Fig. 1 CO₂/Carbonate promotion effects in 2e⁻-WOR. a** Schematic illustration of $CO_2$/carbonate promoted electrochemical $H_2O$ oxidation to $H_2O_2$. With the mediator effects of $CO_2$/carbonate, the WOR reaction pathway could change from 4e⁻ towards $O_2$ to 2e⁻ towards $H_2O_2$. **b** $H_2O_2$ selectivity in Ar-saturated and $CO_2$-saturated 1.0 M sodium phosphate buffer solution. $CO_2$ serves as a promoter for 2e⁻-WOR, as the $H_2O_2$ selectivity increased by orders of magnitude compared to that without $CO_2$. **c–e** I–V curves, $H_2O_2$ FEs, and $H_2O_2$ partial current densities in 1.0 M $NaHCO_3$ and 1.0 M $Na_2CO_3$, respectively. The maximum $H_2O_2$ FE in 1.0 M $NaHCO_3$ was 34% with a $H_2O_2$ partial current density of 5.6 mA cm⁻². In comparison, the maximum $H_2O_2$ FE in 1.0 M $Na_2CO_3$ was 56%, and the maximum $H_2O_2$ partial current density was 68 mA cm⁻².

together with $H_2O_2$ are close to 100%, suggesting no significant side reactions in this electrochemical system (Fig. 2e, see Methods). In addition, FTO electrodes with different thicknesses, fluorine doping levels, or surface resistivity exhibited quite similar $H_2O_2$ selectivity (Supplementary Fig. 6b), further confirming that the dominant factor on WOR pathway is from carbonate concentrations. Other types of common conducting metal oxides, including alumina-doped zinc oxide and indium tin oxide, were also evaluated in carbonate electrolyte but showed poor stability under the WOR conditions (Supplementary Fig. 2). Titanium mesh, which is an OER inert metal catalyst, also exhibited good 2e⁻-WOR performance in 1.0 M $Na_2CO_3$ with a maximum $H_2O_2$ FE of ~ 47% at 2.36 V (Supplementary Fig. 2e, f). However, it presented poor stability due to possible surface passivation under high oxidation potentials (Supplementary Fig. 2 g). These above experimental results strongly support our hypothesis that $CO_2$/carbonate may directly participate in and promote the $H_2O_2$ generation process as a promising 2e⁻-WOR mediator.

**Electrochemical $H_2O_2$ generation at industrial-relevant current densities.** To further amplify the carbonate mediation effect for improved $H_2O_2$ generation performance, we evaluated the electrochemical 2e⁻-WOR performance of FTO electrode in high concentration carbonate solutions. Figure 3a, b and Supplementary Fig. 7 show the I–V curves and corresponding $H_2O_2$ FEs under different potentials. In 2.0 M $Na_2CO_3$ solution, we achieved high $H_2O_2$ FEs of ~60 to 70%, while delivering large current densities of up to 800 mA cm⁻². This impressive $H_2O_2$ performance can be even further improved by using 5.0 M of $K_2CO_3$. The reason why we chose to use $K_2CO_3$ is due to its higher solubility in water (112.3 g in 100 g water at 25 °C) than that of $Na_2CO_3$ (29.4 g in 100 g water at 25 °C)[39]. The FTO catalyst achieved a 10 mA cm⁻² onset current density at 2.75 V vs. RHE in 5.0 M $K_2CO_3$, which is 50 mV lower than that in 2.0 M $Na_2CO_3$. With the overpotentials gradually increased, the $H_2O_2$

FE quickly ramped up to a plateau of over 80% under a wide range of current densities of up to 1 A cm⁻² (Fig. 3b). We achieved a maximal $H_2O_2$ FE of 87% at 600 mA cm⁻², representing a 522 mA cm⁻² $H_2O_2$ partial current density (Fig. 3b). More impressively, at 3.7 V, the catalyst reached a current density of 1 A cm⁻² while still maintaining a high $H_2O_2$ selectivity of 78%, achieving an industrial-relevant $H_2O_2$ partial current density of 780 mA cm⁻². The $H_2O_2$ generation rate can be further boosted to a maximal partial current of 1.3 A cm⁻² (73% FE at 1.8 A cm⁻² overall current) using an extended-range power supply. This corresponds to an unprecedented $H_2O_2$ production rate of 24.3 mmol cm⁻² h⁻¹ and is orders of magnitude higher compared to previous reports (Fig. 3c, d). Please be noted here that this power supply was not equipped with a three-electrode system therefore the anode potential could not be accurately measured (see Methods). Such high electrochemical $H_2O_2$ generation rates benefit from the sufficient mass diffusions in aqueous solutions where no triple-phase boundary is needed, which shows an advantage compared to the cathodic $H_2O_2$ generation from 2e⁻-ORR (oxygen reduction reaction) where $O_2$ gas diffusions typically limit its reaction rates to hundreds of milliamps per square centimeter (Supplementary Note 2)[2,40]. Finally, long-stability is usually a big challenge for 2e⁻-WOR due to the high oxidation potentials that could damage the electrode[9,11,16,17,41,42] (Supplementary Table 1). Besides, in traditional batch reactors, as the products continuous to accumulate during the long-term stability test (Supplementary Fig. 8a), the electrolyte environment will be changed and thus the electrocatalytic performance continuously decayed. Our FTO catalyst electrode, coupling with a flow reactor (Supplementary Fig. 8b), presented excellent durability under water oxidation conditions, maintaining a stable potential and high $H_2O_2$ FEs of over 80% to deliver a 150 mA cm⁻² current for 250 hours (Fig. 3e and Supplementary Table 1).

The good selectivity, activity, and stability of our carbonate mediated 2e⁻-WOR makes it possible for this anode reaction to be coupled with 2e⁻-ORR cathodic reaction to double the efficiency of

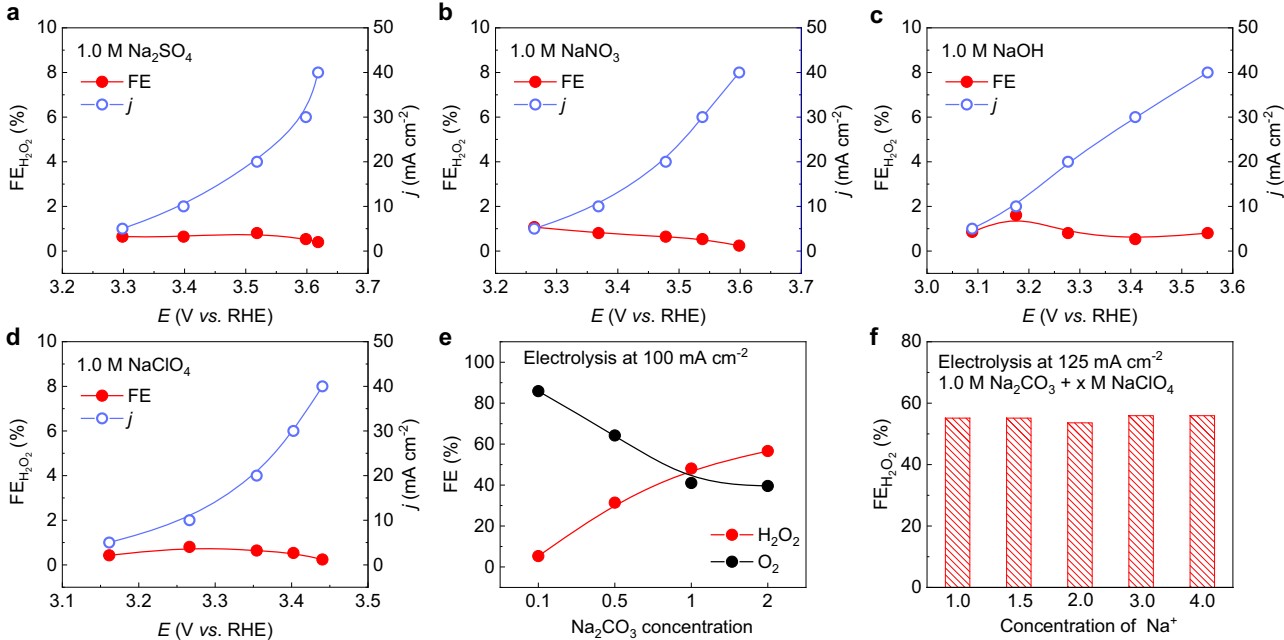

**Fig. 2 Impacts of anion species and anion/cation concentrations on 2e⁻-WOR. a–d** I–V curves and $H_2O_2$ FEs of FTO catalyst using 1.0 M $Na_2SO_4$, 1.0 M $NaNO_3$, 1.0 M NaOH, and 1.0 M $NaClO_4$, respectively. The $H_2O_2$ FEs in these electrolytes were lower than 2%, indicating there are no promotion effects for $H_2O_2$ formation using these anion species. **e** Dependence of $H_2O_2$ and $O_2$ selectivity on $Na_2CO_3$ concentration at 100 mA cm⁻². The $H_2O_2$ FE was increased with increased $Na_2CO_3$ concentration, while the FE of side product $O_2$ was correspondingly decreased, indicating that $Na_2CO_3$ could be directly involved in the 2e⁻-WOR process. **f** $H_2O_2$ FE in 1.0 M $Na_2CO_3$ with different concentrations of $NaClO_4$.

electrons in producing $H_2O_2$ from both electrodes (Supplementary Fig. 9a). On the anode side, $H_2O$ can be oxidized to $H_2O_2$ via carbonate mediation by our high-performance 2e⁻-WOR catalyst. On the cathode side, we used oxidized carbon black (demonstrated in our previous study[40]) as the selective 2e⁻-ORR catalyst to reduce $O_2$ into $H_2O_2$ (Supplementary Fig. 9b, c)[40,43]. Based on the previous definition of $H_2O_2$ FE on one side of the electrode, the maximal overall $H_2O_2$ FE in this two-electrode system is 200%. As shown in Supplementary Fig. 9d, our system delivered a 100-mA cell current (1 cm² FTO electrode) at 2.5 V cell voltage with a high overall $H_2O_2$ FE of 140%, suggesting a significant improvement compared to either 2e⁻-WOR or 2e⁻-ORR system. Furthermore, to fully use generated $H_2O_2$ and $Na_2CO_3$ mediator, we designed a process (Supplementary Fig. 9a) for a continuous generation of an adduct product between $Na_2CO_3$ and $H_2O_2$ ($Na_2CO_3 \cdot 1.5H_2O_2$, Supplementary Fig. 9e–g)[11,44].

**Mechanism studies**. To gain a molecular level understanding of the reaction mechanism in our $CO_2$/carbonate mediated 2e⁻-WOR, we employed molecular dynamics simulations, coupled with experimental studies, to reveal the most possible reaction pathway. Reaction intermediates that could exist in electrolyte under the applied high oxidation potentials (~ 3 V vs. RHE)[45], including $CO_3^{•-}$, $OH^•$, and percarbonates ($HCO_4^-$ or $C_2O_6^{2-}$), were taken into consideration[34,46]. We first proposed several possible reaction pathways as summarized in Supplementary Fig. 10. After an initial screening based on theoretical studies and experimental observations, we proposed that the carbonate-mediated water oxidation to $H_2O_2$ could proceed via the following four reaction intermediate steps with most favorable thermodynamics[7,28,47,48] $CO_3^{•-}$, $HCO_4^-$, $HCO_3^- + H_2O_2$, and $CO_2 + H_2O_2 + OH^-$. We used ab initio molecular dynamics (AIMD) to evaluate the thermodynamics of these intermediate steps. A 12 Å × 12 Å × 12 Å cubic supercell with 57 water molecules was used to simulate bulk water and maintain a density

of 1 g cm⁻³ (Supplementary Fig.11a, b). To assess the energies of these intermediates in ionic form in aqueous solution, seven $H_2O$ molecules were replaced with $CO_3^{•-}$ or $HCO_4^-$ in succession (Fig. 4a and Supplementary Fig. 11b, c), while $HCO_3^- + H_2O_2$ or $CO_2 + H_2O_2 + OH^-$ replaced eight $H_2O$ molecules (Supplementary Fig. 11d, e). Taking the simulation of $CO_3^{•-}$ as an example here, we first evaluated the convergence of the average energy. Figure 4a showed running average of total energy in different averaging time windows from 0.5 ps to 2.0 ps. We found that the more stable average energy is obtained at 2.0 ps window because the fluctuations in average energy between positive and negative were less than 0.1 eV. Therefore, we took the average energy of the last 2.0 ps as the energy of $CO_3^{•-}$ in Supplementary Table 2, which also applies to the other three intermediates. According to the AIMD simulation results, here we suggest a carbonate-mediated 2e⁻ water oxidation reaction mechanism as shown in Fig. 4b (all these elementary reactions are exothermic under the electrochemical potential range we operated): First, $CO_3^{2-}$ is oxidized to $CO_3^{•-}$ at high oxidation potentials on the electrode surface ($CO_3^{2-} \rightarrow CO_3^{•-} + e^-$, $\Delta G = -0.72$ eV at U = 3.0 V vs. RHE). Second, the generated $CO_3^{•-}$ coupled with a $H_2O$ molecule is further oxidized to $HCO_4^-$ ($CO_3^{•-} + H_2O \rightarrow HCO_4^- + H^+ + e^-$, $\Delta G = -0.27$ eV at U = 3.0 V vs. RHE). Subsequently, $HCO_4^-$ is hydrolyzed to generate $H_2O_2$ and converted back to $HCO_3^-$ or $CO_2$ ($HCO_4^- + H_2O \rightarrow HCO_3^- + H_2O_2$ or $HCO_4^- + H_2O \rightarrow CO_2 + H_2O_2 + OH^-$, $\Delta G = -0.31/0.42$ eV without applied potential). As shown in Supplementary Fig. 12, the first and second elementary reactions are electrochemical steps. Under an electrode potential of 3.0 V, both reactions are exothermic. The third fundamental reaction is a non-electrochemical step, and the calculation results show that the former is more likely to occur than the latter. During this reaction step, $H^+$ in $H_2O$ combines with the –OOH group in $HCO_4^-$ molecule to generate $H_2O_2$, and the remaining $OH^-$ group from $H_2O$ can combine with remaining $CO_2$ from $HCO_4^-$ to form $HCO_3^-$. We note here that this hydrolysis step involves O

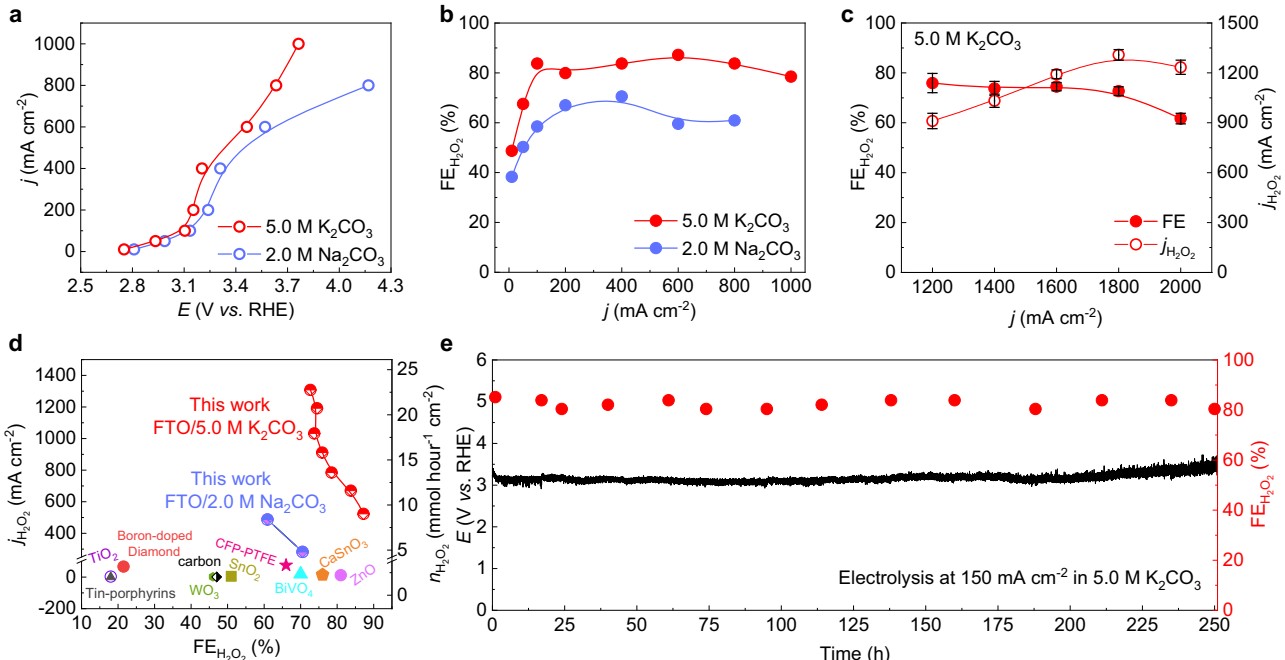

**Fig. 3 Electrochemical 2e⁻-WOR performance using high concentration carbonate mediator. a** I–V curves of FTO catalyst in 2.0 M $Na_2CO_3$ and 5.0 M $K_2CO_3$. $K_2CO_3$ was chosen due to its higher solubility than $Na_2CO_3$. The maximal current density in our standard three-electrode cell was cut at 1 A cm⁻². **b** The corresponding $H_2O_2$ FEs. The highest FE reached 87% in 5.0 M $K_2CO_3$ with a current density of 600 mA cm⁻² under 3.46 V. **c** $H_2O_2$ FEs and partial currents in 5.0 M $K_2CO_3$ solution with current densities greater than 1 A cm⁻². The error bars represent two independent tests. **d** Comparison of $H_2O_2$ selectivity and activity between this work and previous reports[7,9,11–14,41]. **e** The stability test of FTO catalyst by maintaining a 150 mA cm⁻² WOR current in 5.0 M $K_2CO_3$ solution. Its $H_2O_2$ selectivity and potential remained stable for a 250-hour continuous operation.

exchange between water and carbonate ions, which becomes an important evidence to be confirmed by experiments. Finally, $HCO_3^-$ or $CO_2 + OH^-$ generated from the anode, coupled with the remaining $OH^-$ group generated from the cathode, is concerted back to $CO_3^{2-}$ to close the mediation loop in the system (Supplementary Fig. 13). We also found out that the FTO electrode provides a conducting and stable surface to extract electrons from carbonate, and may not serve as a classic catalytic surface where proper chemical bonds with reaction intermediates were usually established to facilitate reaction steps (Supplementary Figs. 14 and 15).

To support the proposed reaction mechanism, we first tested the formation of carbonate radical by electron paramagnetic resonance (EPR). Due to the short lifetime of carbonate radical, we used 5,5-dimethyl-1-pyrroline N-oxide (DMPO) as a spin trap and an in-situ trapping method (see Methods) to detect its generation under electrolysis (Supplementary Fig. 16a)[49]. Compared to the direct mixture of DMPO and before/post-electrolysis carbonate solution, the solution obtained from the in-situ trapping method under electrolysis exhibit a clear four-line 1:2:2:1 splitting pattern characteristic of the DMPO•−OH adduct (Fig. 4c and Supplementary Fig. 16a), indicating that carbonate radical was formed at the surface of FTO electrode under oxidation potentials. Furthermore, we performed an ¹⁸O isotope labeling experiment to support the possible formation of $HCO_4^-$ intermediates. As we mentioned earlier, the hydrolysis process of $HCO_4^-$, if existing in the electrolyte, will lead to oxygen exchange between carbonate and water (Supplementary Fig. 16b). Therefore, if we use ¹⁸O isotope-labeled water as the electrolyte, we would expect an increased abundance of ¹⁸O (δ¹⁸O) in $Na_2CO_3$ after electrolysis compared to that of natural exchange without electrolysis (Methods)[11]. Two sets of ¹⁸O isotope experiments were therefore designed: First, to figure out

the natural exchange rate as the background, we tested δ¹⁸O as a function of time in $Na_2CO_3$ after it was dissolved in ¹⁸O labeled water; Next, we did the same tests of δ¹⁸O at the same time points under WOR electrolysis. As clearly shown in Fig. 4d, the abundance of ¹⁸O isotope in $Na_2CO_3$ under electrolysis condition continues to increase over time, and is several orders of magnitude higher than that without electrolysis, suggesting the violent interaction and chemical bond reconfiguration between carbonate and water via the percarbonate intermediate pathway, instead of a direct water oxidation pathway (Supplementary Fig. 17).

In conclusion, we demonstrated a $CO_2$/carbonate mediated electrochemical water oxidation for high-performance $H_2O_2$ generation, where carbonate ions help to steer the 4e⁻ reaction pathway to 2e⁻ via intermediates such as carbonate radicals and percarbonate, and FTO electrode provides a highly conductive, stable, and 4e⁻-OER inert surface. As a result, we achieved a high $H_2O_2$ selectivity of up to 87%, industrial-relevant $H_2O_2$ generation partial current of up to 1.3 A cm⁻², as well as excellent stability, suggesting a promising route for the renewable and onsite generation of $H_2O_2$ using electricity. This mediation process could be further extended to other electrochemical oxidation applications, especially for those suffering from 4e⁻-OER competition such as chlorine evolution, hydrocarbon oxidation, nitrogen oxidation, etc. Future works can be focused on exploring other catalytic materials enabling this mediation reaction, improving the onset potentials, and further extending the durability.

## Methods
**Materials**. FTO was purchased from MSE Supplies (2.2 mm 7-8 Ohm/Sq FTO TEC 7 Coated Glass Substrates, used in all experiments if there is no other noting) and Sigma (SKU: 735159 and 735256). AZO and ITO were purchased from Sigma. Titanium mesh was purchased from Kunshan GuangJiaYuan new materials Co.

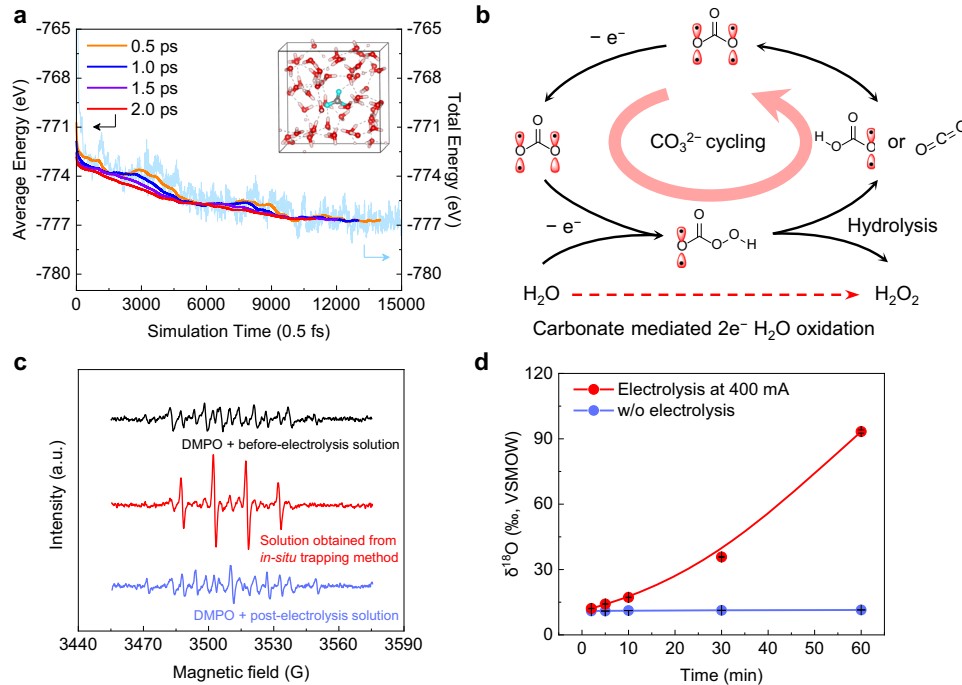

**Fig. 4 Molecular understanding of the reaction mechanism and experimental detection of key intermediates. a** Convergence of average energy of $CO_3^{\bullet-}$ with 50 $H_2O$ molecules using AIMD. Evaluating average of total energy was done over a varying length of averaging time, from 0.5 ps to 2.0 ps. The time coordinate for the average energy corresponds to the point from which the averaging window begins (e.g., for an averaging window of 0.5 ps, the average that is shown at 500 fs corresponds to the average from 500 fs to 1,000 fs). Cyan, red, light pink, and gray balls represent the O in $CO_3^{\bullet-}$, the O in $H_2O$, H, and C atoms, respectively. **b** Reaction mechanism of carbonate-mediated $2e^-$-WOR to $H_2O_2$. Two possible key intermediates, $CO_3^{\bullet-}$ and $HCO_4^-$, were proposed, to facilitate the $2e^-$ pathway. **c** EPR spectra of $Na_2CO_3$ electrolyte containing DMPO spin trap. Compared to the solutions of non-electrolysis or post-electrolysis, the solution obtained from the in-situ trapping method under electrolysis exhibit a clear four-line 1:2:2:1 splitting pattern characteristic of the DMPO$^\bullet$ − OH adduct, indicating the formation of carbonate radical intermediates under electrolysis. **d** $^{18}O$ isotope abundance in dried carbonate (Methods) at varied isotope exchange time between $Na_2CO_3$ solute and deionized water without or without electrolysis. The $^{18}O$ isotope abundance in dried carbonate under WOR conditions was orders of magnitude higher than that of natural exchange background, indicating the C-O bonding dissociation and formation due to the percarbonate hydrolysis step. The error bars represent two independent tests.

Ltd. The oxidized carbon was prepared according to a previous report[40]. Hydrophobic carbon fiber paper (CFP, PTFE loading 60$wt$.%) was purchased from Fuel Cell store. Chemicals were purchased from Sigma and VWR International Company.

**Electrocatalytic oxidation of $H_2O$[11].** The electrochemical measurements were run at 25 °C in a customized gas-tight H-type glass cell separated by Nafion 117 membrane (Fuel Cell Store). A BioLogic VMP3 workstation was employed to record the electrochemical response. In a typical three-electrode system, a platinum foil (Beantown Chemical, 99.99%) and a saturated calomel electrode (SCE, CH Instruments) were used as the counter and reference electrode, respectively. The FTO electrodes were used as the working electrodes, and stainless-steel alligator clip was used to connect the FTO glass electrode. The alligator clip was not immersed in the electrolyte. The geometric area of the FTO electrode was 0.5 to 1 cm², which was precisely defined by an electrochemically inert, hydrophobic wax (Apiezon wax WW100) during electrochemical tests. Before electrochemical measurements, all samples were pre-stabilized at 20 mA cm⁻² to clean the FTO surface. All potentials measured against SCE ($E_{SCE}$) were converted to the RHE ($E_{RHE}$) scale in this work using $E_{RHE} = E_{SCE} + 0.244\,V + 0.0591 \times pH$, where pH values of the electrolytes were determined by an Orion 320 PerpHecT LogR Meter (Thermo Scientific). The pH values of investigated electrolytes are listed in Supplementary Table 2. The electrolyte in the anodic compartment was stirred at a rate of 1,600 r.p.m. during electrolysis. All the measured potentials were manually compensated unless stated otherwise. The overall resistance was determined by potentiostatic electrochemical impedance spectroscopy at frequencies ranging from 0.1 Hz to 200 kHz, and manually compensated as $E$ (iR corrected versus RHE, where iR is the voltage drop from overall resistance) = $E$ (versus RHE) – R (the overall resistance, including the electrolyte resistance and the intrinsic resistance of the electrode) × i (amps of average current). The impedance number was about 12 to 17 Ω. DC stabilized power supply (ITECH) was used for the ultra-high current density (over 1 A cm⁻²) electrolysis. The volume of the electrolyte solution was 25 mL, and the reaction time was determined by the electrolysis current to insured the total electrolysis coulomb was about 10 to 50 C. After electrolysis, the generated $H_2O_2$ was detected by using the standard potassium permanganate (0.1 N KMnO₄

solution, Sigma-Aldrich) titration process, according to the following equation:

$$2MnO_4^- + 5H_2O_2 + 6H^+ \rightarrow 2Mn^{2+} + 5O_2 + 8H_2O$$

The typical quantification time is about 5~10 min, which was much longer than the lifetime of carbonate radical (the lifetime of radical is about several microseconds)[50].

In this work, sulfuric acid (2.0 N $H_2SO_4$, VWR International Company) was used as the $H^+$ source. The FE for $H_2O_2$ production is calculated using the following equation:

$$FE = \frac{\text{generated } H_2O_2 \text{ (mol)} \times 2 \times 96485 \text{ (C mol}^{-1})}{\text{total amount of charge passed (C)}} \times 100 \text{ (maximum 100\%)}$$

For the stability test, a continuous three-electrodes flow cell was employed to continuously produce $H_2O_2$ (Supplementary Fig. 8b). The electrolyte in the anodic compartment was stirred at a rate of 1600 r.p.m. during electrolysis. A peristaltic pump (Longer) was used to pump in $K_2CO_3$ electrolyte to the WOR side, and another peristaltic pump was used to pump out generated $H_2O_2$ in the WOR side. The electrolyte flow rate of the WOR side was 6 mL min⁻¹. The FE for $H_2O_2$ production is calculated using the following equation:

$$FE = \frac{\text{generated } H_2O_2 \text{ (molL}^{-1}) \times 2 \times 96485 \text{ (C mol}^{-1}) \times \text{flow rate (mL s}^{-1})}{j_{total}(mA)}$$
$$\times 100 \text{ (maximum 100\%)}$$

For the $2e^-$-WOR//$2e^-$-ORR electrosynthetic cell test, 0.5 mg cm⁻² oxidized carbon catalyst was air-brushed onto 2-cm² Sigracet 35 BC gas diffusion layer (Fuel Cell Store) electrodes as $2e^-$-ORR cathode. Then, a 1-cm² FTO electrode was used as anode. The two electrodes were therefore placed on opposite sides in the H cell. $O_2$-saturated 2.0 M $Na_2CO_3$ was used as electrolyte. The cathode was open to the atmosphere. The flow rate of 2.0 M $Na_2CO_3$ electrolyte was 3 ml min⁻¹ at both sides, as controlled by a peristaltic pump. A current of 100 mA was employed for $H_2O_2$ production. The FE of the electrosynthetic cell for $H_2O_2$ production is calculated using the following equations, respectively:

$$FE = \frac{\text{generated } H_2O_2 \text{ (mol L}^{-1}) \times 2 \times 96485 \text{ (C mol}^{-1}) \times \text{flow rate (mL s}^{-1})}{j_{total}(mA)} \times 100 \text{ (maximum 200\%)}$$

To obtain solid $H_2O_2$, the electrolyte after electrolysis (two-electrode configuration) was firstly concentrated by rotary evaporator, then about 100 ml of absolute isopropanol is added into 20 ml of the electrolyte for extraction, and the mixture is mechanically stirred. The precipitate is separated by vacuum filtration and washed several times with absolute isopropanol. The isolated precipitate is then dried in a vacuum oven at room temperature for 24 h.

To quantify the gas products during electrolysis, argon gas (Airgas, 99.995%) was delivered into the anodic compartment at a rate of 50.0 standard cubic centimeters per minute (sccm, monitored by an Alicat Scientific mass flow controller) and vented into a gas chromatograph (Shimadzu gas chromatography-2014) equipped with a combination of molecular sieve 5 Å, Hayesep Q, Hayesep T and Hayesep N columns. A thermal conductivity detector was mainly used to quantify gas product concentration. The partial current density for the $O_2$ produced was calculated as follows:

$$j_i = x_i \times v \times \frac{n_i F p_o}{RT} \times (electrode\ area)^{-1}$$

where $x_i$ is the volume fraction of certain product determined by online GC referenced to calibration curves from the standard gas sample (Airgas), $v$ is the flow rate of 50.0 sccm, $n_i$ is the number of electrons involved, $p_o = 101.3$ kPa, F is the Faradaic constant, T= 298 K and R is the gas constant. The corresponding Faradaic efficiency at each potential is calculated by FE = $\frac{j_i}{j_{total}} \times 100\%$. Of note, $H_2O_2$ FEs and $O_2$ FEs are evaluated from two completely different quantification systems: one is from solution titration, and another is from gas chromatographic method. When adding together, as shown in Fig. 2e, the total FEs for $H_2O_2$ and $O_2$ are close to 100% (within testing error range), suggesting there are no side reactions other than WOR under the electrolysis conditions.

**Characterizations.** SEM was performed on a FEI Quanta 400 field emission scanning electron microscope. Powder X-ray diffraction data were collected using a Bruker D2 Phaser diffractometer in parallel beam geometry employing Cu Kα radiation (λ = 1.54056 Å) and a 1-dimensional LYNXEYE detector, at a scan speed of 0.02° per step and a holding time of 1 s per step. X-ray photoelectron spectroscopy was obtained with a PHI Quantera spectrometer, using a monochromatic Al Kα radiation (1486.6 eV) and a low energy flood gun as neutralizer. All XPS spectra were calibrated by shifting the detected carbon C 1 s peak to 284.6 eV.

**[18]O isotope measurement.** We used [18]O isotope-labeled water to prepare the $Na_2CO_3$ electrolyte. First, 0.14 mol (14.8 g) $Na_2CO_3$ (Sigma) was dissolved into 70 ml of ultrapure Milli-Q water. Then, 0.6 ml 10% $H_2O^{18}$ was added. After stirring for 5 min, the fresh 2.0 M $Na_2CO_3$ solution was used as the electrolyte for water oxidation. After electrolysis at 400 mA for a certain time, 1 ml of the electrolyzed 2.0 M $Na_2CO_3$ solution was taken out; meanwhile another 1 ml solution was taken from the bulk 2.0 M $Na_2CO_3$ solution without electrolysis. Then both of the sample solutions were quickly precipitated by 5 ml 1.0 M $BaCl_2$. Next, the solution was centrifuged and washed with ultrapure Milli-Q water. The powder was then dried at 60 °C for [18]O isotope analysis. Of note, the natural exchange time with the water was the same for the unelectrolyzed and electrolyzed $Na_2CO_3$. Oxygen isotope analysis was performed on a Gas Bench-Conflo-Isotope Ratio Mass Spectrometer (Delta V, Thermo Scientific) system. Then ~0.2 mg of sample powder was weighed out into an exetainer, which was flushed by helium flow for 10 min on the GasBench. Then 0.3 ml of 105% phosphoric acid was added into the exetainer for 24-h reaction. The generated $CO_2$ gas was then delivered to the mass spectrometry for isotope ratio analysis. The oxygen isotope composition of the carbonate was calculated based on the measured oxygen isotope composition of the $CO_2$ gas, based on the fractionation factor between the two at the reaction temperature. The isotope experiments to monitor the [18]O isotope offers an analytical precision (1σ) of 0.05‰ for δ[18]O and the values are reported as standard δ notation with respect to the Vienna Standard Mean Ocean Water (VSMOW)[51].

**EPR measurement.** Electrochemical generated carbonate radical was performed in 1.0 M $Na_2CO_3$ solution with a constant electrolysis current of 200 mA. Once the potential became stable, pipette contains DMPO solution (200 mM, 100 μl) was immersed into the electrolyte and make sure the pipette tip was fully contact with the FTO electrode. The radical generated on the electrode will in-situ react with the DMPO solution in the interface between pipette and FTO electrode, and the final solution (sample named solution obtained from in-situ trapping method) contains 100 μl electrolyte and 100 μl DMPO solution. For the sample named DMPO + $Na_2CO_3$ solution, DMPO solution (200 mM, 100 μl) was directly mixed with 1.0 M $Na_2CO_3$ (100 μl). For the sample named DMPO + post-electrolysis solution, DMPO solution (200 mM, 100 μl) was mixed with the electrolyte (100 μl) after electrolysis at 200 mA for 5 min. Finally, above obtained solutions were transferred to capillary for EPR test. All EPR measurements were taken at room temperature with EPR spectrometer (Bruker A300). We set typical parameters as follows: 3515 G center of field; 9.852054 GHz frequency; 1 G modulation amplitude; 40 msec conversion time; 81.92 msec time constant; 20.38 Mw microwave power; 120 G scan width; 40 s sweep time.

**Molecular dynamic calculations.** A cubic box 12 Å × 12 Å × 12 Å was modeled with 57 water molecules to maintain a density of water at ~1 g cm$^{-3}$ (Supplementary Fig. 11b). To fit the box of this size, $CO_3^{\bullet-}$, $HCO_4^-$, $HCO_3^- + H_2O_2$ and $CO_2 + H_2O_2 + OH^-$ replaced seven, seven, eight, and eight water molecules, respectively. The negative charge ($^-$) of the ion was neutralized by a uniform background charge. We did not consider the hydroxide ions and protons in the electrolyte during simulation. Total energy of the above four possible intermediates was calculated by AIMD at a constant temperature of 300 K (using Nose-Hoover thermostat[52], with a time step of 0.5 fs) as implemented in the Vienna ab initio simulation program (VASP)[53,54]. Perdew-Burke-Ernzerhof (PBE) functional[55] and DFT-D3[56] methods were used to describe the exchange and correlation energies and the van der Waals interactions, respectively. A plane-wave cutoff energy of 400 eV and Gamma centered k-mesh of 1 × 1 × 1 were set in MD simulations. Considering the calculation cost, we ran 7.5 ps for each structure. From these AIMD calculations, we analyzed the arithmetic average of their total energies using different averaging time windows from 0.5 ps to 2.0 ps. It is found that averaging within a time window of 2.0 ps gives a value fluctuates around the final value by ±0.1 eV, less than other time windows. Therefore, average energy was evaluated from last 2.0 ps AIMD trajectory, as shown in Supplementary Table 3. The formation of $CO_3^{\bullet-}$ and $HCO_4^-$ are electrochemical oxidation steps. Supplementary Table 4 shows inorganic standard electrode potentials[45]. The reaction between $H_2O$ and $CO_3^{\bullet-}$ was calculated by two separated process including $H_2O \rightarrow OH^\bullet + H^+ + e^-$ and $CO_3^{\bullet-} + OH^\bullet \rightarrow HCO_4^-$ because the energy of $H^+ + e^-$ item in $CO_3^{\bullet-} + H_2O \rightarrow HCO_4^- + H^+ + e^-$ reaction step is difficult to accurately calculate by MD simulations. Therefore, we deal with this step by adding the above two steps together. To better match the results of the experiment, we convert the standard electrode potential into a reversible hydrogen electrode ($U_{RHE} = U_{SHE} + 0.059 \times$ pH, pH = 12). Take $CO_3^{2-} \rightarrow CO_3^{\bullet-} + e^-$ reaction as an example, the standard electrode potential for this reaction is +1.57 ± 0.03 V and the potential corresponding to conversion to RHE is 2.278 V (pH=12, $U_{RHE} = +1.57 + 0.059 \times 12 = 2.278$ V), corresponding to the reaction free energy of 0 eV. When the electrode potential is increased to 3 V, the reaction is exothermic with an energy of −0.72 eV.

**Gibbs free energies on the $SnO_2$(110) surface calculations.** The exchange-correlation potential is described by the generalized gradient approximation (GGA) with spin polarized revised Perdew-Burke-Ernzerhof (RPBE)[57] functional due to that it is more adept at describing chemisorption on metals. The projector augmented wave is applied to describe the electron-ion interaction and the plane-wave energy cutoff is set to 400 eV. All structures are optimized with a convergence criterion of 1 × 10$^{-5}$ eV for the energy and 0.02 eV/Å for the forces. The $SnO_2$(110) surface is the most common of the faces of rutile structure tin oxide (Space group: P42/mnm, No. 136)[58]. Therefore, the $SnO_2$(110) surface is modeled using a slab with a (4 × 2) surface supercell consisting of 3 trilayers and containing 144 atoms. The vacuum spacing is set to more than 15 Å for surface isolation to prevent interaction between two neighboring surfaces. The top two trilayers are fully relaxed during the structural optimization and geometry optimizations for $SnO_2$(110) are performed with 1 × 1 × 1 k-mesh. The Poisson-Boltzmann (PB) implicit solvation model, Vaspsol[59], was used to describe the effect of solvation as implemented in VASP 5.4.4, with a dielectric constant ε = 80 for water. All molecules were performed in a 15 × 15 × 15 Å$^3$ unit cell with a 5 × 5 × 5 k-point grid sampling.

Computational hydrogen electrode (CHE)[60] model was used to calculate the Gibbs free energy change for OER elementary reactions. At electrode potential U = 0 V (vs. reversible hydrogen electrode, RHE), the Gibbs free energy change (ΔG) can be calculated by

$$\Delta G = \Delta E + \Delta E_{ZPE} - T\Delta S + \Delta G_U$$

where ΔE is the energy difference between the products and reactants from DFT computations; $\Delta E_{ZPE}$ and ΔS are the changes in zero-point energy and entropy, respectively, which are obtained from the vibrational frequency calculations; T is the temperature at 298 K. The energy corrections of gas-phase species in this work, including zero point energies (ZPE) and entropies (TS), are listed in Supplementary Table 5. $\Delta G_U = -neU$, where U is the electrode applied potential relative to RHE, $e$ is the elementary charge transferred and $n$ is the number of proton–electron pairs transferred. 2e$^-$-WOR path is calculated in alkaline condition. We calculated the chemical potential of hydroxides and electrons with reference to previous work[61]. μ(e$^-$)− μ(OH$^-$) − $eU$ = 9.81 eV was obtained by repeated calculations under reversible hydrogen electrode potential ($U_0$(RHE) =1.23 V) at T = 298.15 K.

The accuracy of the above DFT methodology applied in our work is mainly based on published work. More details can be found in the Supporting Information.

## Data availability

The data that support the findings of this study are available from the corresponding authors upon reasonable request.

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

## Acknowledgements

This work was supported by Rice University, the Robert A. Welch Foundation (grant no. C-2051-20200401), the David and Lucile Packard Foundation (grant no. 2020-71371), and the Alfred P. Sloan Foundation (grant no. FG-2021-15638). C.X. acknowledges support from a J. Evans Attwell-Welch postdoctoral fellowship provided by the Smalley-Curl Institute. L.F. acknowledges support from the China Scholarship Council (CSC) (201806320253) and 2018 Zhejiang University Academic Award for Outstanding Doctoral Candidates. Yuanyue Liu acknowledges the support by NSF (Grant No. 1900039), ACS PRF (60934-DNI6), and the Welch Foundation (Grant No. F-1959-20210327). The calculations used computational resources at XSEDE, TACC, Argonne National Lab, and Brookhaven National Lab. X.B. acknowledges support from the China Scholarship Council (CSC) (201906090150).

## Author contributions

L.F., X.B. and C.X. contributed equally. L.F. and H.W. conceived the project and designed the experiments. H.W. supervised the project. L.F., C.X., Xiao Zhang, Y.X., and Z.W. perform the experimental study. X.B., Xunhua Zhao, and Yuanyue Liu performed the theoretical study. L.F., X.B., Yingying Lu, Yuanyue Liu, and H.W. wrote the manuscript with support from all authors.

## Competing interests

The authors declare no competing interests.
