## [Peer Review File · Nature Communications]

CO₂/carbonate-mediated electrochemical water oxidation to hydrogen peroxideREVIEWER COMMENTS

Reviewer #1 (Remarks to the Author):

The authors reported the H₂O₂ production from carbonate solution on F-doped SnO₂ conductive glass (FTO) electrode with high current density and faraday efficiency. The current of 1300 to 2000 mA/cm² on flat FTO is amazing. The purchased FTO (7-8 Ohm/Sq, TEC 7) is very common. Assuming that the average length from the reaction point to the stainless-steel alligator clip is 1cm, if FTO area was 1x1cm, this large current density and the resistance from the Fig. 1 and Fig. 3 are abnormal considering all component resistances with FTO, electrolyte solution, and surface reaction overpotential for water oxidation. It is difficult to show the steep slope of I-V on 7-8 Ohm/Sq of FTO.

In Fig. 1c, the slope of I-V at 2.8 -3.1V was changed at 3.1 – 3.3 V-RHE. Why this increase happened?

The dependences of I-V and FE-V between Na₂CO₃ and NaHCO₃ are very different. The difference between CO₃²⁻ and HCO₃⁻ are originated from pH. Many H⁺ is formed on FTO during the water oxidation to H₂O₂, and the partial pH on the FTO surface become acid, and CO₃²⁻ is changed to HCO₃⁻ and H₂CO₃ (CO₂). It means the difference between CO₃²⁻ and HCO₃⁻ are minimized at high current density. But, why the results were opposite?

If CO₃²⁻ is main species for the reaction, how about the data in Na₂CO₃ + NaOH?

The dependence of pH in wider range should be measured. The main species, CO₃²⁻ or HCO₃⁻, can be clear. The properties at 2.7 – 3.1 V in Na₂CO₃ and NaHCO₃ are similar. The mechanism was changed at 3.1 V?

The descriptions on the reactor and reaction conditions are not enough. The volume of electrolyte solution and reaction time (coulomb), except stability and O₁₈ experiments, are not clear.

They mentioned the image of industrial process in Supplementary Fig. 4. The concentration of H₂O₂ (mol/L) is very important factor, and the time dependence of H₂O₂ concentration should be added. If volume of electrolyte solution is very large or the reaction time is very short, the impact of faraday efficiency of H₂O₂ is not high. There are many papers of high faraday efficiency of H₂O₂ nearly 100% at low concentration.

The oxidation of H₂O₂ to O₂ is very serious problem. Why this oxidation was inhibited on simple FTO in this experiment?

Reviewer #2 (Remarks to the Author):

The authors present an interesting report on H₂O₂ production from FTO mediated through carbonate. My main concern is on statements of novelty that I feel are very, and unnecessarily, oversold. The concept of carbonate enhancing H₂O₂ formation has been reported, as has the mechanism shown in Figure 4. That being said, the quality of the research is high and the article is still interesting, since enhancing the production at FTO from very little to very efficient is a very interesting concept that has not, to my knowledge, been shown before. The weighting and the focus of the article should be carefully rewritten to better show what is novel from the authors and what is building on existing literature.

Line 47 - The authors state in their introduction that current WOR catalysts are limited to ~10 mA cm⁻² partial current densities. However published examples have reported partial current densities much

higher than this (CaSnO₃ ~ 90 mA cm⁻² {<https://doi.org/10.1002/adfm.202100099>}, BDD ~200 mA cm⁻² {<https://doi.org/10.1021/acseenergylett.1c00904>}, PTFE/CF ~70 mA cm⁻² {<https://doi.org/10.1038/s41929-019-0402-8>}). This is not to take away from the high performance catalyst reported here, but this value should be corrected.

Line 62 - The authors link biological ROS reactions enhancement of the WOR to H₂O₂, which implies that they are the first to conceive of this concept. This is not the case, since multiple publications have reported reactions of H₂O₂ with CO₂ in the concept of ECL (<https://doi.org/10.1021/acs.analchem.0c04212>, <https://doi.org/10.1021/jacs.9b11842>) peroxy carbonate synthesis (<https://doi.org/10.1016/j.elecom.2003.11.014>) and have already reported mechanistic links between CO₂ and enhancing WOR to H₂O₂ (<https://doi.org/10.1021/acssuschemeng.0c07263>, <https://doi.org/10.1002/cssc.201900560>). Similarly, the mechanism shown in Figure 4 of H₂O₂ formation via HCO₄⁻ was previously proposed for photooxidations on BiVO₄/WO₃ catalysts (<https://doi.org/10.1039/C6CC01605G>)

Line 148 - Repeated sentence

Line 210 - The combined reactor uses the FTO anode presented here (FE=87%) and the reference oxidised carbon black cathode (FE~90%) but the reactor selectivity (140%) was much less than the sum of these. Could the authors add a comment to why a combined reactor does not achieve the sum of its parts, and how further work could address this?

Response to reviewer's comments:

We appreciate the reviewers for their constructive comments, which have helped us significantly improve our research and the quality of our manuscript. We have now included additional experiments and explanation to fully address the reviewers' concerns and suggestions. Below, we have addressed the points raised by reviewers one by one.

REVIEWER COMMENTS

Reviewer #1

The authors reported the H₂O₂ production from carbonate solution on F-doped SnO₂ conductive glass (FTO) electrode with high current density and faraday efficiency. The current of 1300 to 2000 mA/cm² on flat FTO is amazing. The purchased FTO (7-8 Ohm/Sq, TEC 7) is very common. Assuming that the average length from the reaction point to the stainless-steel alligator clip is 1cm, if FTO area was 1x1cm, this large current density and the resistance from the Fig. 1 and Fig. 3 are abnormal considering all component resistances with FTO, electrolyte solution, and surface reaction overpotential for water oxidation. It is difficult to show the steep slope of I-V on 7-8 Ohm/Sq of FTO.

Response

We highly appreciate the reviewer's important suggestions/comments that have greatly helped us to improve the quality and depth of our work. As widely adapted in electrochemistry community, to exclude the influence from the ohmic drop in the reactor and electrode, we always need to do iR-compensation to obtain the intrinsic potential of the electrochemical reaction. We have manually corrected all the potentials with iR-compensation, where R includes the working electrode resistance and electrolyte resistance, and i is the current at a particular applied potential (DOI: 10.1016/j.mtphys.2020.100253). We have added an addition explanation in the experimental section, also copied below.

All the measured potentials were manually compensated unless stated otherwise. The overall resistance was determined by potentiostatic electrochemical impedance spectroscopy at frequencies ranging from 0.1 Hz to 200 kHz, and manually compensated as E (iR corrected versus RHE, where iR is the voltage drop from overall resistance) = E (versus RHE) - R (the overall resistance, including the electrolyte resistance and the intrinsic resistance of the electrode) \times i (amps of average current).

Comment 1

In Fig. 1c, the slope of I-V at 2.8 -3.1V was changed at 3.1 – 3.3 V-RHE. Why this increase happened?

Response 1

Thanks for your question. As we know, there are two possible reaction pathways of water oxidation reaction (WOR). The total current density was influenced by both two-electron WOR and four-electron WOR. At 2.8-3.1 V vs. RHE, the main reaction pathway was four-electron WOR towards oxygen, as shown in Fig. R1a-b (Fig. 1c-d), the H_2O_2 FE was lower than 40%. When the applied potential increased more than 3.1 V vs. RHE, the main reaction pathway moved to two-electron pathway, and the H_2O_2 FE was about 60%. The different reaction pathways could cause the changing of the slope of I-V curve. Besides, 3.1 V is a sufficiently potential for carbonate mediated water oxidation, the heterogeneous electrochemical process can be driven fast enough that only mass transfer controls the current (Bard, A. J.; Faulkner, L. R. *Electrochemical Methods: Fundamentals and Applications*, 2nd ed.; Wiley: New York, 2001). The high concentration carbonate and fast stirring rate (1600 rpm) could satisfy the mass transfer. Taken together, the current density increased quickly after 3.1 V. Such phenomenon also happened in other electrochemical reactions, such as CO_2 reduction (DOI: 10.1126/science.abg6582) and two-electron oxygen reduction (DOI: 10.1038/s41467-021-24329-9), as shown in Fig. R1c-d.

Fig. R1 | a-b, I-V curves and H₂O₂ FEs in 1.0 M NaHCO₃ and 1.0 M Na₂CO₃, respectively. **c**, I-V curve of CO₂RR on Cu in acid electrolyte, adapted from DOI: 10.1126/science.abg6582. **d**, I-V curve of O₂ reduction reaction on different carbon catalyst in KOH, adapted from DOI: 10.1038/s41467-021-24329-9.

Comment 2

The dependences of I-V and FE-V between Na₂CO₃ and NaHCO₃ are very different. The difference between CO₃²⁻ and HCO₃⁻ are originated from pH. Many H⁺ is formed on FTO during the water oxidation to H₂O₂, and the partial pH on the FTO surface become acid, and CO₃²⁻ is changed to HCO₃⁻ and H₂CO₃ (CO₂). It means the difference between CO₃²⁻ and HCO₃⁻ are minimized at high current density. But, why the results were opposite?

If CO₃²⁻ is main species for the reaction, how about the data in Na₂CO₃ + NaOH?

Response 2

Thanks for raising this important point, which serves important side evidence to support our proposed working mechanism. As we concluded in our manuscript that the carbonate, not the bicarbonate or CO₂, is the key mediator for H₂O₂ generation via a carbonate radical reaction pathway. As shown in Fig. R2 (Fig. 4b) and Fig. R3 (Supplementary Fig. 6), the initial step in carbonate solutions involves the carbonate oxidation to carbonate radical, during which no protons were produced on the surface of the electrode. The second step involves the generation of protons, but since the key intermediate percarbonate has been formed, the locally generated protons won't impact the following generation of H₂O₂. The electrolyte was also violently stirred to ensure a good mass transport for minimized local pH change. However, in bicarbonate solutions, the surface reaction is dominated by water oxidation with protons produced starting from the initial electron transfer step, different from the case of carbonate and thus presents different reaction pathways.

Fig. R2 | Reaction mechanism of carbonate-mediated 2e⁻-WOR to H₂O₂. Two possible key intermediates, CO₃^{•-} and HCO₄⁻, were proposed, to facilitate the 2e⁻ pathway.

Fig. R3. | Screening of possible reaction pathways. Based on possible species existing in the electrolyte under the oxidation potentials we tested, we proposed several possible reaction pathway as listed in (a-g). **a**, H_2O is first oxidized to OH^\cdot , and then two OH^\cdot coupled to form H_2O_2 . If H_2O_2 generated from this pathway, then the anions would not have a direct impact on the H_2O_2 selectivity. However, as shown in our experimental results, we found out that carbonate ions would significantly change the WOR pathway. Additionally, the standard electrochemical potential of H_2O -to- OH^\cdot conversion is $+3.438 \pm 0.017 \text{ V vs. RHE}$, which is much higher than the onset potentials ($\sim 2.75 \text{ V vs. RHE}$ for FTO electrode and 2.1 V vs. RHE for titanium mesh) we detected H_2O_2 generation. As a result, we excluded this pathway, as well as pathways listed in **b-d**, involving the generation of OH^\cdot . **e**, In this proposed pathway, CO_3^{2-} is first oxidized to $\text{CO}_3^{\cdot-}$, and then two $\text{CO}_3^{\cdot-}$ coupled together to form $\text{C}_2\text{O}_6^{2-}$. The standard electrochemical potential of $\text{CO}_3^{\cdot-}$ formation from CO_3^{2-} oxidation is $+2.278 \pm 0.03 \text{ V vs. RHE}$, it's close to the onset potentials for H_2O_2 generation on FTO and titanium mesh. However, the following $\text{CO}_3^{\cdot-}$ coupling step is difficult due to the strong repulsion between two negatively charged ions. Therefore, this step is excluded. **f**, This mechanism also excluded due to the strong repulsion between two negatively charged ions ($\text{CO}_3^{\cdot-}$ and OH^-) during their coupling. Finally, we focused on pathways in (g) and (h) for further calculations.

We have also tested the electrochemical water oxidation performance in $\text{Na}_2\text{CO}_3 + \text{NaOH}$ according to the reviewer's suggestion. As shown in Fig. R4 (also copied in Supplementary Fig. 3c), the H_2O_2 selectivity decreased with the increased NaOH concentration. In alkaline electrolytes, adsorbed OH^- only serves as the intermediate of oxygen evolution reaction (DOI: 10.1039/D0SC01532F), which cannot promote carbonate-mediated two-electron water oxidation. However, OH^- absorption will decrease the coverage ratio of carbonate on FTO surface. As a result, the FE of H_2O_2 decreased with the increment of NaOH concentration. These results further supported our conclusion that carbonate plays a key role in the selectivity changing from O_2 to

H₂O₂. We have added additional discussions to discuss this result on Page 5 in the revised manuscript, also copied below.

Fig. R4 | H₂O₂ FE in 1.0 M Na₂CO₃ with different concentrations of NaOH.

To further confirm the carbonate promotion effects, we then tested the electrochemical water oxidation performance in Na₂CO₃ + NaOH electrolyte. In alkaline electrolytes, OH⁻ absorption plays a key role in oxygen evolution reaction. With the increase of NaOH concentration, OH⁻ absorption will decrease the coverage ratio of carbonate absorption on FTO surface, and thus leading to decreased H₂O₂ selectivity and increased OER, which further demonstrates that high pH is not the reason for high H₂O₂ selectivity when compared to bicarbonate solutions (Supplementary Fig. 3c).

Comment 3

The dependence of pH in wider range should be measured. The main species, CO₃²⁻ or HCO₃⁻, can be clear. The properties at 2.7 – 3.1 V in Na₂CO₃ and NaHCO₃ are similar. The mechanism was changed at 3.1 V?

Response 3

Thanks for your comments. Our studies focus on investigating carbonate-mediated two-electron electrochemical water oxidation. According to CO₂/bicarbonate/carbonate thermodynamics, carbonate will react with CO₂ or H⁺ and form HCO₃⁻ if we shift the carbonate electrolyte into a lower pH bound. Even though it is impossible to independently change the pH and carbonate concentration for a decoupled study of pH and carbonate effect, we did the following experiments to exclude the pH effect and demonstrate that, the higher H₂O₂ selectivity under a more concentrated carbonate solution is due to the higher carbonate concentration, but not due to the associated pH increase. We compared the electrochemical water oxidation reaction performance in 1.0 M NaOH (pH = 14.0) and 5.0 M K₂CO₃ (pH = 13.2), which exhibit similar pH value. We found the electrolyte pH value has no relationship with H₂O₂ selectivity, as shown in Fig. R5. We believe the abovementioned explanation and comparison could exclude the possible effects from electrolyte pH.

Fig. R5 | Control experiments to exclude possible pH effects. 1.0 M NaOH exhibit an even higher pH value than 5.0 K₂CO₃. However, the H₂O₂ selectivity in 1.0 M NaOH was much lower than in 5.0 M K₂CO₃.

The electrochemical properties at 2.7-3.1 V vs. RHE in Na₂CO₃ and NaHCO₃ are similar, which may be due to the chemical equilibrium between carbonate and bicarbonate will generate a certain concentration of carbonate in the NaHCO₃ electrolyte. As shown in R6, the current density at 2.7-3.1 V vs. RHE is low (< 20 mA cm⁻²), which means only a trace amount of carbonate was needed to mediate the two-electron water oxidation reaction. As a result, at 2.7 -3.1 V vs. RHE, the carbonate concentration in both bicarbonate and carbonate can satisfy the mediation process. So the electrochemical performance at 2.7-3.1 V in Na₂CO₃ and NaHCO₃ are similar. At higher applied potentials (> 3.1 V vs. RHE), the electrochemical H₂O oxidation can be driven fast enough that only mass transfer controls the current. The Na₂CO₃ electrolyte has enough carbonate mass transport to deliver high current densities. However, the NaHCO₃ electrolyte does not have enough carbonate species because of the chemical equilibrium, which result in the partial current density of H₂O₂ in 1.0 M NaHCO₃ was only 5 mA cm⁻² at higher applied potentials.

Fig. R6 | Electrochemical performance in 1.0 M Na₂CO₃ and 1.0 M NaHCO₃. a-c, I-V curves, H₂O₂ FEs, and H₂O₂ partial current densities in 1.0 M NaHCO₃ and 1.0 M Na₂CO₃, respectively. The maximum H₂O₂ FE in 1.0 M NaHCO₃ was 34% with a H₂O₂ partial current density of 5.6 mA cm⁻². In comparison, the maximum H₂O₂ FE in 1.0 M Na₂CO₃ was 56%, and the maximum H₂O₂ partial current density was 68 mA cm⁻².

Comment 4

The descriptions on the reactor and reaction conditions are not enough. The volume of electrolyte solution and reaction time (coulomb), except stability and O18 experiments, are not clear.

Response 4

Thanks for your comments. We have added additional descriptions in the experimental section to make the reaction conditions more clearly, also copied below.

The volume of the electrolyte solution was 25 mL, and the reaction time was determined by the electrolysis current to insured the total electrolysis coulomb was about 10 to 50 C.

Comment 5

They mentioned the image of industrial process in Supplementary Fig. 4. The concentration of H₂O₂ (mol/L) is very important factor, and the time dependence of H₂O₂ concentration should be added. If volume of electrolyte solution is very large or the reaction time is very short, the impact of faraday efficiency of H₂O₂ is not high. There are many papers of high faraday efficiency of H₂O₂ nearly 100% at low concentration.

Comment 6

The oxidation of H₂O₂ to O₂ is very serious problem. Why this oxidation was inhibited on simple FTO in this experiment?

Response 5 & 6

We believe these two comments are related so we combine them together in this response.

We appreciate the reviewer's important suggestions. We have added the time dependence of H₂O₂ concentration in the revised Supplementary Fig. 4 and Fig. R7a. As shown in Fig. R5a, the concentration of H₂O₂ increased with the increase of electrolysis time, and we can get about 600 ppm high concentration H₂O₂ after 60 minutes electrolysis (the volume of the electrolyte is 25 mL).

We agree with the reviewer that the oxidation of H₂O₂ to O₂ is very serious problem because H₂O₂ is unstable and could be oxidized to O₂ in oxidation environments or at a higher temperature. Compared with other materials, such as platinum and other metals, FTO exhibits inert catalytic activity. However, the H₂O₂ oxidation cannot be fully inhibited by electrode materials design. To solve this problem, we developed a three-electrode flow cell for continuous H₂O₂ generation (Fig. R7b and Supplementary Fig. 4b). The electrolyte continuously pumps into the flow reactor, and H₂O₂ can be continuously generated. The flow reactor design can also avoid further H₂O₂ oxidation. The production rate over the 250 hours was about 26.4 $\mu\text{mol min}^{-1}$, and we can continuously produce 150 ppm (4.4 mM) H₂O₂ solution over 250 hours. We have added additional discussions on Page 9 in the revised manuscript, also copied below.

Besides, in traditional batch reactors, as the products continuous to accumulate during the long-term stability test (Supplementary Fig. 4a), the electrolyte environment will be changed and thus the electrocatalytic performance continuously decayed. Our FTO catalyst electrode, coupling

with a flow reactor (Supplementary Fig. 4b), presented excellent durability under water oxidation conditions, maintaining a stable potential and high H_2O_2 FEs of over 80% to deliver a 150 mA cm^{-2} current for 250 hours (Fig. 3e and Supplementary Table 1).

Fig. R7 | Stability test of $2e^-$ -WOR. **a**, Time dependence of H_2O_2 concentration in traditional batch cell. **b**, Schematic illustration of a continuous flow reactor for long-term stability test. Our continuous flow reactor resolves the H_2O_2 accumulation challenge by maintaining a stable electrocatalysis environment for long term operation.

Besides, we also developed a process to transform generated H_2O_2 into $\text{Na}_2\text{CO}_3 \cdot 1.5\text{H}_2\text{O}_2$, as shown in Fig. R8 (Fig. S4). Compare with H_2O_2 solution, $\text{Na}_2\text{CO}_3 \cdot 1.5\text{H}_2\text{O}_2$ is more stable and more accessible to transport and storage.

Fig. R8 | Practical application of 2e⁻-WOR. **a**, Schematic illustration of our H₂O₂ generation from both electrode by coupling 2e⁻-WOR and 2e⁻-ORR, together with the preparation of an adduct product between Na₂CO₃ and H₂O₂ (Na₂CO₃·1.5H₂O₂). After electrolysis, the electrolyte solution was concentrated and separated to increase the H₂O₂ concentration in the Na₂CO₃ solution. The high concentration H₂O₂-Na₂CO₃ solution was then directly extracted to get pure solid Na₂CO₃·1.5H₂O₂ powder. The right picture showed the degradation of KMnO₄ using as-obtained solid Na₂CO₃·1.5H₂O₂ powder. **b,c**, I-V curve and corresponding H₂O₂ FE_s of 2e⁻-ORR using oxidized carbon catalyst in 2.0 M Na₂CO₃. **d**, Cell voltage and H₂O₂ FE of our 2e⁻-ORR//2e⁻-WOR cell as a function of time by fixing a cell current at 100 mA. **e**, Chemical structure of Na₂CO₃·1.5H₂O₂. **f**, The XRD pattern for as-extracted Na₂CO₃·1.5H₂O₂ from electrolyte after electrolysis. Inset is the photo of as obtained Na₂CO₃·1.5H₂O₂. **g**, O 1s XPS spectrum of Na₂CO₃·1.5H₂O₂. The solid-state Na₂CO₃·1.5H₂O₂ can avoid the storage and transportation challenges of liquid-phase H₂O₂ solution because liquid-phase H₂O₂ solutions will happen self-accelerating decomposition reactions if there are any contaminants. In addition, liquid-phase H₂O₂ solutions exhibit high leakage risk during storage and transportation.

Reviewer #2

The authors present an interesting report on H₂O₂ production from FTO mediated through carbonate. My main concern is on statements of novelty that I feel are very, and unnecessarily, oversold. The concept of carbonate enhancing H₂O₂ formation has been reported, as has the mechanism shown in Figure 4. That being said, the quality of the research is high and the article is still interesting, since enhancing the production at FTO from very little to very efficient is a very interesting concept that has not, to my knowledge, been shown before. The weighting and the focus of the article should be carefully rewritten to better show what is novel from the authors and what is building on existing literature.

Response

We appreciate the reviewer's positive comments. In the present version of the manuscript, we have addressed all questions and concerns raised by the reviewer and added additional explanation to better show the novelty of our work. Furthermore, according to the reviewers' suggestions, we have carefully checked the literature regarding 2e⁻-WOR and included related ones in the revised manuscript.

We apologize for the impression of an "oversold novelty" that we might brought to the reviewer, which has now been carefully examined and rewritten to avoid any confusions or misunderstanding. The novelty of our work comes from two major aspects. **First**, coupling advanced molecular dynamic simulations with electron paramagnetic resonance and isotope labeling experiments, to our best knowledge, our work is the first one to systematically and with solid evidence chain to reveal the carbonate-mediation process and reaction mechanism of electrochemical H₂O oxidation to H₂O₂. Our MD simulations and *in situ* experiments suggested that carbonate mediates the WOR pathway to H₂O₂ by forming carbonate radical and percarbonate intermediates. None of these systematic mechanism studies and solid evidences have been provided in previous literature. We completely agree that there were several hypotheses proposed before but none of them have been experimentally and systematically validated. **Second**, after revealing and a full understanding of this carbonate mediation effect, **we made a full use of it to achieve unprecedented H₂O₂ performances** in terms of all catalysis metrics including selectivity, activity, and stability. We achieved high H₂O₂ selectivity of up to 87% under 600 mA cm⁻² (the highest H₂O₂ selectivity in the literature was 81% under ~ 15 mA cm⁻², DOI: 10.1021/acscatal.8b04873), delivered ultrahigh H₂O₂ partial current densities up to 1.3 A cm⁻² (the highest H₂O₂ partial current density in the literature was ~ 200 mA cm⁻² in the literature, DOI: 10.1021/acseenergylett.1c00904), achieved an excellent stability of a stable and continuous H₂O₂ generation for 250 hours with over 80% H₂O₂ selectivity at 150 mA cm⁻² current density (the longest stability test in the literature was 72 hours with only 0.7 mA cm⁻² current density, DOI: 10.1021/acscatal.8b04873). The electrochemical performance of our work represents significant improvements compared to the existing literature. We find the electrochemical performances as well as the systematic mechanism studies to be significant strengths of this work.

Comment 1

Line 47 - The authors state in their introduction that current WOR catalysts are limited to ~10 mA cm⁻² partial current densities. However published examples have reported partial current densities much higher than this (CaSnO₃ ~ 90 mA cm⁻² {<https://doi.org/10.1002/adfm.202100099>}, BDD ~200 mA cm⁻² {<https://doi.org/10.1021/acseenergylett.1c00904>}, PTFE/CF ~70 mA cm⁻² {<https://doi.org/10.1038/s41929-019-0402-8>}). This is not to take away from the high performance catalyst reported here, but this value should be corrected.

Response 1

We highly appreciate this important advice and apologize for not stating the current literature status clearly. We have cited these papers as Ref. 11, 15, 16 in our work. We have now revised our description in our introduction part accordingly, also copied below.

These catalysts (typically made of inert metal oxides⁷⁻¹⁰ as well as other materials¹²⁻¹⁴) requires large overpotentials to activate the water oxidation step, but their 2e⁻-WOR current densities are usually limited at ~ 10 to 200 mA cm⁻², as the extra overpotentials to drive larger currents would start to push the water oxidation reaction all the way down to O₂ with significantly decreased H₂O₂ selectivity^{7-11,15,16}.

Comment 2

Line 62 - The authors link biological ROS reactions enhancement of the WOR to H₂O₂, which implies that they are the first to conceive of this concept. This is not the case, since multiple publications have reported reactions of H₂O₂ with CO₂ in the concept of ECL (<https://doi.org/10.1021/acs.analchem.0c04212>, <https://doi.org/10.1021/jacs.9b11842>) peroxycarbonate synthesis (<https://doi.org/10.1016/j.elecom.2003.11.014>) and have already reported mechanistic links between CO₂ and enhancing WOR to H₂O₂ (<https://doi.org/10.1021/acssuschemeng.0c07263>, <https://doi.org/10.1002/cssc.201900560>). Similarly, the mechanism shown in Figure 4 of H₂O₂ formation via HCO₄⁻ was previously proposed for photooxidations on BiVO₄/WO₃ catalysts (<https://doi.org/10.1039/C6CC01605G>)

Response 2

We appreciate the reviewer's important suggestions and apologize for giving the reviewer the impression of claiming the first to conceive of this concept. We have now carefully revised the description and cited related references as Ref. 29-34 in our revised work to avoid any confusions, also copied below.

Inspired by this phenomenon and previous studies in electrogenerated chemiluminescence^{29,30} and peroxycarbonate synthesis³¹, here we hypothesize that CO₂ (or

carbonate as its ionic form in water) may serve as an effective mediator to promote the H_2O_2 pathway in electrochemical water oxidation (Fig. 1a)^{32–34}.

Comment 3

Line 148 - Repeated sentence

Response 3

Thanks for your carefully reading. We are sorry for this mistake. We have deleted the repeated sentence.

Comment 4

Lien 210 - The combined reactor uses the FTO anode presented here (FE=87%) and the reference oxidised carbon black cathode (FE~90%) but the reactor selectivity (140%) was much less than the sum of these. Could the authors add a comment to why a combined reactor does not achieve the sum of its parts, and how further work could address this?

Response 4

Thanks for your question. We tested the combined reactor using FTO anode and oxidized carbon black cathode in 2.0 M Na_2CO_3 , and the cell current is 100 mA. The FE of H_2O_2 at the cathode side was about 90%. The geometric surface area of FTO anode was 1 cm^2 . As shown in Fig. R9 (Fig. 3b), the FE of H_2O_2 was 58% at 100 mA cm^{-2} in 2.0 M Na_2CO_3 . As a result, the sum H_2O_2 FE of the anode part and cathode part was close to the combined reactor. Future work should focus on reactor design to further promote the ORR activity and thus increase the total H_2O_2 FEs at high current densities.

Fig. R9 | H_2O_2 FEs of FTO catalyst in 2.0 M Na_2CO_3 .

REVIEWER COMMENTS

Reviewer #1 (Remarks to the Author):

I understood that most of data were corrected by iR-compensation. But the corrected data are often far from the original data. The iR drop of FTO is very large, and it is intrinsic point in this work. Author should add the original data without iR-compensation especially about Fig. 1, 2, and 3.

Reviewer #2 (Remarks to the Author):

The authors have provided a highly detailed and comprehensive rebuttal giving additional weight to an already strong paper. I am happy to recommend this article for publication.

Reviewer #3 (Remarks to the Author):

The authors present a detailed study on a catalytic process to convert water to hydrogen peroxide and propose a carbonate mediated scheme. Their study focuses on fluorine doped tin oxide and includes a thorough electrochemical analysis including isotope labeling experiments and a computational study (DFT and AIMD) on the proposed mechanism. Furthermore the authors even demonstrate a practical application for the proposed process in the framework of an electrochemical peroxide H₂O₂ production.

It seems that in the first revision of the manuscript, the previous referees mostly had questions on the experimental, mechanistic and electrochemical parts as well as the cited literature. From the response, it seems the authors have made a thorough effort to address all comments accordingly. They have extended explanations, added new data and included further references. I am also happy to see that the authors have taken the criticism of overselling seriously and revised the manuscript thoroughly.

While my expertise in the experimental part is limited, my main focus lies on the computational part, which seems thoroughly done using state of the art methods. Modelling using explicit solvent and AIMD is an important aspect here and the authors present a technically sound simulation and analysis.

So overall, the paper can be published in Nature Communications with minor revision, as the reproducibility of the computed results is suboptimal and the given details on the simulation results are insufficient.

a) Complete free energy diagrams for all investigated mechanisms should be given and a comment on the role of barriers should be made. Furthermore, structures or at least structures for snapshots should be supplied in the SI that allow to reproduce the simulated systems.

b) For the inclusion of the electrochemical potential, the authors use a somewhat outdated a-posteriori approach and use experimental reference potentials to shift the computed values. This way, whether a step is "electrochemical" or not is an assumption that is made prior to the calculation and introduces a bias of the results. Furthermore, the simulation of a unit cell of a certain system might exhibit a very different Fermi level from what the actual value under electrochemical potential might be. The authors should at least comment on this (see recent work by Goddard, Auer, Neugebauer etc.). This might even be the reason why the authors find unfavorable reaction enthalpies in the surface reaction.

c) In the simulation work, a fairly small unit cell has been applied. The authors should add more information to the SI stating with which hints they have that the simulation results are robust with respect to the size of the simulation cell (which concentrations do they simulate compared to the

experiment ?). Furthermore, a comment on how the pH enters the simulation results should be made.

d) The analysis of the surface reaction on SnO₂ is not perfectly clear - which surface coverage has been assumed ? At an adsorption energy of 3eV one can safely assume that the surface would be fully covered, but which efforts have the authors made to determine the surface termination of their system in experiment and in the simulations ? Which surface termination has been used to obtain the results from Fig. SI 9b ? The provided information is by far too limited to assess the statements given.

minor issues:

e) The authors frequently state that certain experiments have been done "to validate" the mechanism / intermediate / strategy. Note that in science, experiments can only be used to falsify, not to verify or validate (K. Popper). Experiments can support assumptions, but no observation can be made to finally validate a hypothesis. Please rephrase.

f) Please comment on the accuracy of the applied DFT methodology. Which impact would inclusion of exact exchange have and what systematic error do the applied functionals (PBE and PBErev) have.

Response to reviewer's comments:

We appreciate the reviewers for their constructive comments, which have helped us significantly improve our research and the quality of our manuscript. We have now included additional figures and explanation to fully address the reviewers' concerns and suggestions. Below, we have addressed the points raised by reviewers one by one.

REVIEWER COMMENTS

Reviewer #1

I understood that most of data were corrected by iR -compensation. But the corrected data are often far from the original data. The iR drop of FTO is very large, and it is intrinsic point in this work. Author should add the original data without iR -compensation especially about Fig. 1, 2, and 3.

Response

Thanks for your suggestions. We agree with the reviewer's point that the iR -compensated I-V curve usually differs from the original data. The impedance is not only relevant to the FTO's sheet resistance, it is also relevant to the electrolyte resistivity as well as the configuration of the electrochemical device (the distance between working electrode and reference electrode). Therefore, to report the intrinsic catalytic activity, researcher usually report iR -compensated I-V curves to exclude the factor of different reactor, different electrode resistance, different positions of working electrode and reference electrode, etc. The impedance number in our system was about 12 to 17 Ω . According to the reviewer's suggestion, we have now included additional figures to show the original data, as shown in Figs. R1-4 and Supplementary Figs. 3, 4, 5, and 7.

Fig. R1 | Impacts of CO₂ on electrochemical H₂O oxidation. **a**, I-V curves of FTO electrode using CO₂-saturated and Ar-saturated 1.0 M sodium phosphate buffer solution (pH ~ 7). **b**, Corresponding H₂O₂ partial current densities at different potentials. **c-d**, I-V curves and corresponding H₂O₂ FEs of FTO electrode using CO₂-saturated and Ar-saturated 1.0 M sodium phosphate buffer solution (pH ~ 7) without iR compensation.

Fig. R2 | I-V curves and corresponding H₂O₂ FEs in 1.0 M NaHCO₃ and 1.0 M Na₂CO₃ without iR compensation.

Fig. R3 | I-V curves and H₂O₂ FEs of FTO catalyst using 1.0 M Na₂SO₄ (a), 1.0 M NaNO₃ (b), 1.0 M NaOH (c), and 1.0 M NaClO₄ (d) without iR compensation.

Fig. R4 | I-V curves and corresponding H₂O₂ FEs in 2.0 M Na₂CO₃ and 5.0 M K₂CO₃ without iR compensation.

The authors have provided a highly detailed and comprehensive rebuttal giving additional weight to an already strong paper. I am happy to recommend this article for publication.

Response

We thank the reviewer's positive response and support our work for publication.

Reviewer #3

The authors present a detailed study on a catalytic process to convert water to hydrogen peroxide and propose a carbonate mediated scheme. Their study focuses on fluorine doped tin oxide and includes a thorough electrochemical analysis including isotope labeling experiments and a computational study (DFT and AIMD) on the proposed mechanism. Furthermore the authors even demonstrate a practical application for the proposed process in the framework of an electrochemical peroxide H₂O₂ production.

It seems that in the first revision of the manuscript, the previous referees mostly had questions on the experimental, mechanistic and electrochemical parts as well as the cited literature. From the response, it seems the authors have made a thorough effort to address all comments accordingly. They have extended explanations, added new data and included further references. I am also happy to see that the authors have taken the criticism of overselling seriously and revised the manuscript thoroughly.

While my expertise in the experimental part is limited, my main focus lies on the computational part, which seems thoroughly done using state of the art methods. Modelling using explicit solvent and AIMD is an important aspect here and the authors present a technically sound simulation and analysis.

So overall, the paper can be published in Nature Communications with minor revision, as the reproducibility of the computed results is suboptimal and the given details on the simulation results are insufficient.

Response

We thank the referee for the positive remarks and constructive comments to help us further improve the manuscript.

Comment 1

Complete free energy diagrams for all investigated mechanisms should be given and a comment on the role of barriers should be made. Furthermore, structures or at least structures for snapshots should be supplied in the SI that allow to reproduce the simulated systems.

Response 1

Thanks for your suggestions. We have included the complete free energy diagrams corresponding to the reaction mechanisms we studied in Fig. R5 (Supplementary Fig. 13). The first and second elementary reactions are electrochemical steps. Under an electrode potential of 3.0 V, the corresponding free energy barriers are -0.72 and -0.27 eV, respectively, implying

that both reactions are exothermic. The third elementary reaction is a non-electrochemical step, and the free energy barriers for generating $\text{HCO}_3^- + \text{H}_2\text{O}_2$ or $\text{CO}_2 + \text{H}_2\text{O}_2 + \text{OH}^-$ is -0.31 and 0.42 eV, respectively, which means that the former reaction is more likely to occur than the latter. Now following comment on the role of barriers is added in the revised manuscript.

“The first and second elementary reactions are electrochemical steps. Under an electrode potential of 3.0 V, both reactions are exothermic. The third fundamental reaction is a non-electrochemical step, and the calculation results show that the former is more likely to occur than the latter”

In addition, we have provided the coordinates of the snapshot structures in the SI that allow to reproduce the simulated systems.

Fig. R5 | The complete free energy diagrams corresponding to reaction mechanism of carbonate-mediated $2e^-$ -WOR to H_2O_2 .

Comment 2

For the inclusion of the electrochemical potential, the authors use a somewhat outdated a-posteriori approach and use experimental reference potentials to shift the computed values. This way, whether a step is "electrochemical" or not is an assumption that is made prior to the calculation and introduces a bias of the results. Furthermore, the simulation of a unit cell of a certain system might exhibit a very different Fermi level from what the actual value under electrochemical potential might be. The authors should at least comment on this (see recent work by Goddard, Auer, Neugebauer etc.). This might even be the reason why the authors find unfavorable reaction enthalpies in the surface reaction.

Response 2

Thanks for the nice suggestion. We have carefully read the recently published work by Goddard, Auer, Neugebauer etc. Indeed, the Fermi level of the studied system is different from the corresponding actual value under the electrochemical potential, which leads to catalysts with non-zero surface charges that affect the chemical reactivity. In fact, we have been committed to the study of electrochemical reactivity and selectivity under constant potential,

including HER, ORR, CO₂RR and so on. We initially tried to study the reactivity under constant potential (U=3.1 V vs RHE pH=12) as shown in Fig. R6 (Supplementary Fig. 11). However, the second elementary reaction needs to overcome a free energy barrier of 0.33 eV, which is inconsistent with our experimental results, namely, the H₂O₂ FE in 1.0 M Na₂CO₃ is ~45% at ~3.1 V with a H₂O₂ partial current density of 50 mA cm⁻². As a result, we concluded that in our system, H₂O₂ generation reaction may mainly occur in the electrolyte rather than on FTO surface, and FTO only serves as a stable, conducting and inert electrode for the extraction of electrons.

Fig. R6 | Calculations of carbonate adsorption-based mechanism on FTO electrode under constant-potential method. The Fermi level of the studied FTO electrode is different from the corresponding actual value under the electrochemical potential, which leads to FTO electrode with non-zero surface charges that affect the chemical reactivity. We tried to study the reactivity under constant potential (U=3.1 V vs RHE pH=12). However, the second elementary reaction needs to overcome a free energy barrier of 0.33 eV, which is inconsistent with our experimental results.

Comment 3

In the simulation work, a fairly small unit cell has been applied. The authors should add more information to the SI stating with which hints they have that the simulation results are robust with respect to the size of the simulation cell (which concentrations do they simulate compared to the experiment?). Furthermore, a comment on how the pH enters the simulation results should be made.

Response 3

Thanks for the suggestion. Due to the limitation of computational cost, we used a cubic cell with 57 water molecules at density 1 g/cm³ (L = 12 Å). To maintain the density in the aqueous phase close to 1 g/cm³, we removed seven or eight water molecules when adding four reaction intermediates, such as CO₃⁻, HCO₄⁻, HCO₃⁻ + H₂O₂, and CO₂ + H₂O₂ + OH⁻. At this time, the corresponding carbonate concentration is 2%, which is consistent with the experimental carbonate concentration (~1.0 M Na₂CO₃). To demonstrate that the simulation results are robust with respect to the size of the simulation cell, we calculated the arithmetic average of their total energies using different averaging time window from 0.5 ps to 2.0 ps. It is found that averaging within a time window of 2.0 ps gives a value fluctuating around the

final value by ± 0.1 eV. Therefore, the averaging total energy obtained in a 2.0 ps simulation (after allowing the simulation to equilibrate, i.e., locate the local minima) was deemed sufficient. That is, we obtain stable structures and corresponding energies. Now following comment is added supplementary Fig. 12 in the revised SI.

“For all the simulations, we use a cubic cell with 57 water molecules at density 1 g/cm^3 ($L = 12 \text{ \AA}$). To maintain the density in the aqueous phase close to 1 g/cm^3 , we removed seven or eight water molecules when adding four reaction intermediates, such as $\text{CO}_3^{\bullet-}$, HCO_4^- , $\text{HCO}_3^- + \text{H}_2\text{O}_2$, and $\text{CO}_2 + \text{H}_2\text{O}_2 + \text{OH}^-$. At this time, the corresponding carbonate concentration is 2%, which is consistent with the experimental carbonate concentration ($\sim 1.0 \text{ M Na}_2\text{CO}_3$). To demonstrate that the simulation results are robust with respect to the size of the simulation cell, we calculated the arithmetic average of their total energies using different averaging time window from 0.5 ps to 2.0 ps. It is found that averaging within a time window of 2.0 ps gives a value fluctuates around the final value by ± 0.1 eV. Therefore, the averaging total energy obtained in a 2.0 ps simulation (after allowing the simulation to equilibrate, i.e., locate the local minima) was deemed sufficient.”

To better match the results of the experiment, we convert the standard electrode potential into a reversible hydrogen electrode at the experiment pH ($U_{\text{RHE}} = U_{\text{SHE}} + 0.059 \times \text{pH}$, $\text{pH} = 12$). Take $\text{CO}_3^{2-} \rightarrow \text{CO}_3^{\bullet-} + \text{e}^-$ reaction as an example, the standard electrode potential for this reaction is $+1.57 \pm 0.03 \text{ V}$ and the potential corresponding to conversion to RHE is 2.278 V ($\text{pH}=12$, $U_{\text{RHE}} = +1.57 + 0.059 \times 12 = 2.278 \text{ V}$), corresponding to the reaction free energy of 0 eV . When the electrode potential is increased to 3 V , the reaction is exothermic with an energy of -0.72 eV . The calculation of other electrochemical steps is the same. Now following comment on the pH enters is added in the revised manuscript.

“Take $\text{CO}_3^{2-} \rightarrow \text{CO}_3^{\bullet-} + \text{e}^-$ reaction as an example, the standard electrode potential for this reaction is $+1.57 \pm 0.03 \text{ V}$ and the potential corresponding to conversion to RHE is 2.278 V ($\text{pH}=12$, $U_{\text{RHE}} = +1.57 + 0.059 \times 12 = 2.278 \text{ V}$), corresponding to the reaction free energy of 0 eV . When the electrode potential is increased to 3 V , the reaction is exothermic with an energy of -0.72 eV .”

Comment 4

The analysis of the surface reaction on SnO_2 is not perfectly clear - which surface coverage has been assumed? At an adsorption energy of 3 eV on can safely assume that the surface would be fully covered, but which efforts have the authors made to determine the surface termination of their system in experiment and in the simulations? Which surface termination has been used to obtain the results from Fig. SI 9b? The provided information is by far too limited to assess the statements given.

Response 4

Previous experimental and theoretical results showed the preferential growth of SnO_2 films along the $[110]$ in an oxidizing environment and that growth along the $[110]$ direction is

driven by (110) being the lowest energy surface (Please see J. Phys. Chem. C 2014, 118, 11292–11302). In addition, most studies were carried out on SnO₂ (110) surfaces because the (110) surface is the most thermodynamically stable and the most common surface. (see J. Phys. Chem. Lett. 2015, 6, 4224–4228, J. Phys. Chem. C 2014, 118, 28548–28561 and Phys. Rev. B 2002, 65, 245428). Therefore, we chose the (110) surface of SnO₂ to study the surface reaction (For SnO₂ (110) surface, half of the Sn atoms are bare and another half of the Sn atoms are O-terminated). Now following statements are added in the revised SI.

“Since the (110) surface is the most thermodynamically stable surface and has received extensive attention in previous experimental and theoretical studies, we chose the (110) surface of SnO₂ to study the surface reaction (For SnO₂ (110) surface, half of the Sn atoms are bare and another half of the Sn atoms are O-terminated)¹⁻³.”

In experiment, we use 0.5 to 1 cm² FTO glass as the electrode, which is large enough to exclude the surface termination effects. In the simulations, it is assumed that the surface is completely covered by CO₃, because the distance between them is > 4 Å, thus the impact of the degree of coverage on the performance is relatively small. Therefore, to simplify the calculation, we only consider one CO₃ adsorption on the Sn(110) surface for evaluating the oxygen exchange mechanism. Now following statements are added in the revised SI.

“If the surface is completely covered by CO₃, the distance between adjacent CO₃ adsorbates is > 4 Å, thus the impact on performance is relatively small. Here, we only consider one CO₃ adsorption on the Sn(110) surface for evaluating the oxygen exchange mechanism.”

Comment 5

minor issues:

The authors frequently state that certain experiments have been done "to validate" the mechanism / intermediate / strategy. Note that in science, experiments can only be used to falsify, not to verify or validate (K. Popper). Experiments can support assumptions, but no observation can be made to finally validate a hypothesis. Please rephrase.

Response 5

This is a very interesting comment that gives us an opportunity to understand popper's falsification principle. He argued that there is a fundamental logical asymmetry between falsification and verification. Take the case that “swans are all white”, verifying this claim would logically require observing all swans, which is not technologically possible. In contrast, the observation of a single black swan is technologically reasonable and enough to logically falsify the claim, thereby disproving the “swans are all white” conclusion. This is a logical relationship between universal proposition and singular proposition. However, we are now not sure whether our work belongs to universal or singular propositions. Under certain conditions, it may be falsified by practice, and under another condition, it may be validated. We would like to express here that our work is self-consistent, both experimentally and theoretically. We very much hope that our work will receive widespread attention and be replicated in order to

validate or disprove our conclusions. Hope to be able to verify or falsify our work from a philosophical point of view. In the revised manuscript, we have changed “validate” to “support” mechanism/intermediate/strategy.

Comment 6

Please comment on the accuracy of the applied DFT methodology. Which impact would inclusion of exact exchange have and what systematic error do the applied functionals (PBE and PBErev) have.

Response 6

The previous study has shown that the balance between accuracy and computational cost in DFT simulations depends on the choice of exchange and correlation functional (Nat. Commun., 2020, 11, 3509). The accuracy of the DFT methodology applied in our work is based on published work. Taking the SnO₂ (110) surface as an example, RPBE is more accurate for predicting the adsorption energies of small molecules on transition-metal surfaces and does well for those strong chemisorption systems (see Phys. Rev. B, 1999, 59, 7413 and J. Phys. Chem. C, 2017, 121, 4937–4945), and thus the RPBE is chosen for surface studies. When simulating the reactions occurring in electrolytes, strong chemisorption behavior on transition metal surfaces is not involved. Furthermore, it is known that in water, due to the high polarizability of oxygen, vdW interactions have a significant contribution to the binding. The vdW attraction contributes to strengthening both H-bond and non-H-bond interactions, and it increases the overall cohesive energy in the liquid. Schmidt et al. have studied the vdW effect on the density of water with the PBE+dispersion (PBE-D) method (J. Phys. Chem. B 2009, 113, 11959–11964), which includes an interatomic pair potential correction added to the PBE functional. They showed that the density of PBE-D water is very close to the experimental value, and the resulting liquid is also structurally closer to experiments. Therefore, the PBE+D3 is chosen to study the reactions taking place in liquid water, such as Angew. Chem. Int. Ed., 2019, 58, 4210-4216 and J. Chem. Phys., 2016, 144, 130901.

In the inclusion of exact exchange, some proportion of the local exchange-correlation potential is replaced by Hartree-Fock exact-exchange terms, giving very good results for most systems, especially the electrical properties, such as HSE06. However, it's at least an order of magnitude more expensive than GGA calculations. It is very hard to do even a one-step HSE06 calculation at a suitable level of accuracy in the system we studied (more than 150 atoms). Therefore, we used GGA functionals in the calculation process, which can not only provide relatively accurate results, but also obtain reasonable allocation in terms of time and memory.

The previous study has shown that the cause of the systematic errors is the DFT exchange functionals (J. Chem. Theory Comput. 2018, 14, 3083–3090 and Phys. Chem. Chem. Phys., 2009, 11, 1138–1142). The RPBE functional only differs from the PBE functional in the choice of the mathematical form for the exchange energy enhancement factor. The two functionals, PBE and RPBE, follow the same construction logic and therefore contain the same physics and fulfill the same physical criteria (Phys. Rev. B, 1999, 59, 7413). In general, the adsorption energies of small molecules on transition metal surfaces calculated using PBE functional are

larger than that of RPBE. However, the specific error analysis needs to refer to the experimental data. We will try to study this aspect in depth in future work.

REVIEWERS' COMMENTS

Reviewer #3 (Remarks to the Author):

The authors have made a thorough revision of the manuscript and the manuscript can be published after minor revision.

It is somewhat concerning that the constant potential approach is in contradiction with the experimental findings but the corresponding statement of the authors is conclusive. However, there is not detailed description how the calculations for the constant potential approach was carried, as this is a delicate issue in periodic boundary conditions, as adjusting the Fermi level to a physical values sometimes requires charging the unit cell, which requires compensation schemes. Please add a detailed description, otherwise the computed results are not reproducible.

The issues with barriers was raised in the last referee reports and the authors reply that they have included the full free energy profile. However, upon checking the new data in the SI, only total free energies of stable species are found, not transition states have been calculated, no barriers are reported, no computational details about, for example, NEB calculations are given. While this would improve the quality of the calculations in manuscript significantly, as kinetics (and hence activity) is governed by the relative energy of transition states and not minima (unless you assume BEP to be valid, which the authors do not mention), this was not even asked for. It was only asked for discussing this and showing that the authors are aware of the basic assumptions made and the sources of errors in the calculations.

This is also why the falsification argument was brought up - as theory includes so many approximations and neglects so many effects, that it might be used to support some hypothesis, but there are many more possibilities to be right for the wrong reason (see above).

Response 6 is a clear and valid reply, however, it does not help the reader to understand the choice of the authors for their level of theory. Please at least put the detailed reply into the SI and mention it in the main manuscript.

Response to reviewer's comments:

We appreciate the reviewer #3 for their constructive comments, which have helped us significantly improve our research and the quality of our manuscript. We have now included additional explanation to fully address the reviewer's concerns and suggestions. Below, we have addressed the points raised by reviewer #3 one by one.

REVIEWER COMMENTS

Reviewer #3

The authors have made a thorough revision of the manuscript and the manuscript can be published after minor revision.

Response

We thank the reviewer's positive response and support our work for publication.

Comment 1

It is somewhat concerning that the constant potential approach is in contradiction with the experimental findings but the corresponding statement of the authors is conclusive. However, there is not detailed description how the calculations for the constant potential approach was carried, as this is a delicate issue in periodic boundary conditions, as adjusting the Fermi level to a physical values sometimes requires charging the unit cell, which requires compensation schemes. Please add a detailed description, otherwise the computed results are not reproducible.

Response

Thanks for the suggestion. Here we would like to clarify that, the calculations using constant potential method suggest that the reaction is unlikely to occur on the surface, due to the unfavorable thermodynamics for the non-electrochemical (i.e. thermal) step: ${}^*\text{CO}_2\text{-OOH} + \text{H}_2\text{O} \rightarrow {}^*\text{HCO}_3 + \text{H}_2\text{O}_2$. In contrast, the reaction in solution is thermodynamically favorable for all the steps. Thus, we conclude that the reaction is likely to occur in the solution.

The compensation charges are added as point charges in the implicit solvent, following the Poisson-Boltzmann distribution. This has been implemented to VASP as VASPsol (see J. Chem. Phys. 2019, 151, 234101).

Following the reviewer's suggestion, we have added the details about the constant potential calculation into Supplementary Fig. 15:

“The calculations using constant potential method suggest that the reaction is unlikely to occur on the surface, due to the unfavorable thermodynamics for the non-electrochemical (i.e. thermal) step: ${}^*\text{CO}_2\text{-OOH} + \text{H}_2\text{O} \rightarrow {}^*\text{HCO}_3 + \text{H}_2\text{O}_2$. In contrast, the reaction in solution is thermodynamically favorable for all the steps. Thus, we conclude that the reaction is likely to

occur in the solution. For constant potential calculation, we adjust the electron number for every ionic step in order to match the “electrode potential” of the system to the experimental value of 3.1 V vs RHE. For a given structure and electron number, the electron potential Φ with respect to the standard hydrogen electrode (SHE) can be obtained as:

$$\Phi = [E_F - E_{es} - (E_F^{SHE} - E_{es}^{SHE})]/e$$

where E_F is the Fermi level of the system, E_{es} is the electrostatic energy in the middle of implicit solution region, and E_F^{SHE} and E_{es}^{SHE} are the corresponding quantities in SHE. In this work, we use the implicit solvation model as implemented in VASPsol. $E_F^{SHE} - E_{es}^{SHE}$ is benchmarked to be 4.6 eV for VASPsol⁷. Note that the net electronic charges are balanced by the ionic charges in the implicit solution, and thus the total system remains charge neutral.”

Comment 2

The issues with barriers was raised in the last referee reports and the authors reply that they have included the full free energy profile. However, upon checking the new data in the SI, only total free energies of stable species are found, not transition states have been calculated, no barriers are reported, no computational details about, for example, NEB calculations are given.

While this would improve the quality of the calculations in manuscript significantly, as kinetics (and hence activity) is governed by the relative energy of transition states and not minima (unless you assume BEP to be valid, which the authors do not mention), this was not even asked for. It was only asked for discussing this and showing that the authors are aware of the basic assumptions made and the sources of errors in the calculations.

This is also why the falsification argument was brought up - as theory includes so many approximations and neglects so many effects, that it might be used to support some hypothesis, but there are many more possibilities to be right for the wrong reason (see above).

Response

Thanks for the very careful review and detailed comment. We apologize for the confusion. As the reviewer mentioned, we indeed only calculated the thermodynamics. To clarify, we have changed the word “free energy diagrams” to “thermodynamic profile” in Supplementary Fig. 12.

We also agree that the calculation of kinetics would be very valuable but it is computationally extremely expensive for electrochemical interface. In fact, our group has been working on this problem for simple system with well-defined atomic structure (see J. Am. Chem. Soc. 2021, 143, 9423–9428 on ORR and J. Am. Chem. Soc. 2020, 142, 5773–5777 on CO₂RR).

Following the reviewer's suggestions, we have discussed the error sources in Supplementary Fig. 12.

“According to BEP principle, the activation energy is correlated with the reaction transition state energy decreases as the final state energy decreases¹. In our most favorable reaction mechanism, all elementary reaction steps are exothermic in thermodynamics, so we believe that the barriers of their transition states should be small at a potential up to ~3 V. Therefore, the activation energies were not calculated. Such calculations would be very valuable for making more definitive assessment on the reaction mechanism (for example, see our previous works on ORR² and CO₂RR³), although the computational cost is extremely high for simulating the reaction kinetics of electrochemical interface.”

Comment 3

Response 6 is a clear and valid reply, however, it does not help the reader to understand the choice of the authors for their level of theory. Please at least put the detailed reply into the SI and mention it in the main manuscript.

Response

Thanks for the nice suggestion. We have put the detailed response on the accuracy of the DFT method into the revised SI.

“Accuracy of the DFT methodology as applied in our work. The previous study has shown that the balance between accuracy and computational cost in DFT simulations depends on the choice of exchange and correlation functional⁸. Taking the SnO₂ (110) surface as an example, RPBE is more accurate for predicting the adsorption energies of small molecules on transition-metal surfaces and does well for those strong chemisorption systems^{9,10}, and thus the RPBE is chosen for surface studies. When simulating the reactions occurring in electrolytes, strong chemisorption behavior on transition metal surfaces is not involved. Furthermore, it is known that in water, due to the high polarizability of oxygen, vdW interactions have a significant contribution to the binding. The vdW attraction contributes to strengthening both H-bond and non-H-bond interactions, and it increases the overall cohesive energy in the liquid. Schmidt et al. have studied the vdW effect on the density of water with the PBE+dispersion (PBE-D) method¹¹, which includes an interatomic pair potential correction added to the PBE functional. They showed that the density of PBE-D water is very close to the experimental value, and the resulting liquid is also structurally closer to experiments. Therefore, the PBE+D3 is chosen to study the reactions taking place in liquid water.

In the inclusion of exact exchange, some proportion of the local exchange-correlation potential is replaced by Hartree-Fock exact-exchange terms, giving very good results for most systems, especially the electrical properties, such as HSE06. However, it's at least an order of magnitude more expensive than GGA calculations. It is very hard to do even a one-step HSE06 calculation at a suitable level of accuracy in the system we studied (more than 150 atoms).

Therefore, we used GGA functionals in the calculation process, which can not only provide relatively accurate results, but also obtain reasonable allocation in terms of time and memory.

The previous study has shown that the cause of the systematic errors is the DFT exchange functionals^{12,13}. The RPBE functional only differs from the PBE functional in the choice of the mathematical form for the exchange energy enhancement factor. The two functionals, PBE and RPBE, follow the same construction logic and therefore contain the same physics and fulfill the same physical criteria⁹. In general, the adsorption energies of small molecules on transition metal surfaces calculated using PBE functional are larger than that of RPBE. However, the specific error analysis needs to refer to the experimental data. We will try to study this aspect in depth in future work.”

In addition, we have made the following mention in the main manuscript.

“The accuracy of the above DFT methodology applied in our work is mainly based on published work. More details can be found in the Supporting Information.”